



# Quantifying stratospheric biases and identifying their potential sources in subseasonal forecast systems

Zachary D. Lawrence[1,2], Marta Abalos[3], Blanca Ayarzagüena[3], David Barriopedro[3], Amy H. Butler[4], Natalia Calvo[3], Alvaro de la Cámara[3], Andrew Charlton-Perez[5], Daniela I.V. Domeisen[6,7], Etienne Dunn-Sigouin[8], Javier García-Serrano[9], Chaim I. Garfinkel[10], Neil P. Hindley[11], Liwei Jia[12,13], Martin Jucker[14], Alexey Y. Karpechko[15], Hera Kim[16], Andrea L. Lang[17], Simon H. Lee[18], Pu Lin[13,19], Marisol Osman[20,21], Froila M. Palmeiro[9], Judith Perlwitz[2], Inna Polichtchouk[22], Jadwiga H. Richter[23], Chen Schwartz[10], Seok-Woo Son[16], Irina Statnaia[15], Masakazu Taguchi[24], Nicholas L. Tyrrell[15], Corwin J. Wright[11], and Rachel W.-Y. Wu[7]

[1]Cooperative Institute for Research in Environmental Sciences (CIRES), University of Colorado Boulder, Boulder, CO, USA
[2]NOAA Physical Sciences Laboratory (PSL), Boulder, CO, USA
[3]Department of Earth Physics and Astrophysics, Universidad Complutense de Madrid, Madrid, Spain
[4]NOAA Chemical Sciences Laboratory (CSL), Boulder, CO, USA
[5]Department of Meteorology, University of Reading, Reading, UK
[6]University of Lausanne, Lausanne, Switzerland
[7]ETH Zurich, Zurich, Switzerland
[8]NORCE Norwegian Research Centre and Bjerknes Centre for Climate Research, Bergen, Norway
[9]Group of Meteorology, Universitat de Barcelona (UB), Barcelona, Spain
[10]Fredy & Nadine Herrmann Institute of Earth Sciences, The Hebrew University of Jerusalem, Israel
[11]Centre for Space, Atmospheric and Oceanic Science, University of Bath, Bath, UK
[12]University Corporation for Atmospheric Research, Boulder, CO
[13]Geophysical Fluid Dynamics Laboratory, NOAA, Princeton, NJ
[14]Climate Change Research Centre and ARC Centre of Excellence for Climate Extremes, University of8New South Wales, Sydney, Australia
[15]Finnish Meteorological Institute, Meteorological Research, Helsinki, Finland
[16]School of Earth and Environmental Sciences, Seoul National University, South Korea
[17]Department of Atmospheric and Environmental Sciences, University at Albany, State University of New York, Albany, New York
[18]Department of Applied Physics and Applied Mathematics, Columbia University, New York, NY
[19]Program in Atmospheric Science, Princeton University, Princeton, NJ
[20]CONICET – Universidad de Buenos Aires, Centro de Investigaciones del Mar y la Atmósfera (CIMA), Buenos Aires, Argentina
[21]Institute of Meteorology and Climate Research (IMK-TRO), Department Troposphere Research, Karlsruhe Institute of Technology (KIT), Karlsruhe, Germany
[22]European Centre for Medium-Range Weather Forecasts, Reading, UK
[23]Climate and Global Dynamics Laboratory, National Center for Atmospheric Research, Boulder, CO
[24]Department of Earth Science, Aichi University of Education, Kariya, Japan

**Correspondence:** Zachary D. Lawrence (zachary.lawrence@noaa.gov)

**Abstract.** The stratosphere can be a source of predictability for surface weather on timescales of several weeks to months. However, the potential predictive skill gained from stratospheric variability can be limited by biases in the representation of



stratospheric processes and the coupling of the stratosphere with surface climate in forecast systems. This study provides a first systematic identification of model biases in the stratosphere across a wide range of subseasonal forecast systems.

It is found that many of the forecast systems considered exhibit warm global mean temperature biases from the lower to middle stratosphere, too strong/cold wintertime polar vortices, and too cold extratropical upper troposphere/lower stratosphere regions. Furthermore, tropical stratospheric anomalies associated with the Quasi-Biennial Oscillation tend to decay toward each system's climatology with lead time. In the Northern Hemisphere (NH), most systems do not capture the seasonal cycle of extreme vortex event probabilities, with an underestimation of sudden stratospheric warming events and an overestimation

of strong vortex events in January. In the Southern Hemisphere (SH), springtime interannual variability of the polar vortex is generally underestimated, but the timing of the final breakdown of the polar vortex often happens too early in many of the prediction systems.

These stratospheric biases tend to be considerably worse in systems with lower model lid heights. In both hemispheres, most systems with low-top atmospheric models also consistently underestimate the upward wave driving that affects the strength of

the stratospheric polar vortex. We expect that the biases identified here will help guide model development for sub-seasonal to seasonal forecast systems, and further our understanding of the role of the stratosphere for predictive skill in the troposphere.

# 1 Introduction

The Earth's stratosphere is home to several dynamical phenomena that are coupled with the tropospheric circulation. This coupling between the two layers can go in both directions; tropospheric variability drives variability in the stratosphere, but down-

ward coupling from stratospheric variability can subsequently impact weather in the troposphere across the globe (Domeisen and Butler, 2020). As a result, the stratosphere is recognized as a source of predictability for the troposphere on subseasonal-to-seasonal (S2S) timescales (Butler et al., 2019a; Domeisen et al., 2020b). However, model simulations and forecasts often struggle to adequately capture such stratosphere-troposphere coupling processes. Model biases in both the troposphere and stratosphere can impact these coupling processes, with potential deleterious effects for S2S predictability. The goal of this

study is to provide a systematic identification of stratospheric biases in a wide range of S2S forecast systems.

In the wintertime extratropical stratosphere, variability in the upward flux of planetary-scale Rossby waves drives variability in the westerly circulation of the stratospheric polar vortices, affecting their mid-winter strength and the timing of their seasonal breakdowns in spring (Andrews et al., 1987; Garfinkel et al., 2010). During winter and spring, anomalous behavior of the polar vortices can exert an influence on the underlying troposphere. This kind of "downward coupling" is especially apparent for

extreme polar vortex events, including sudden stratospheric warming (SSW) events (Gerber et al., 2012; Baldwin et al., 2021), which are characterized by massive disruptions to the polar vortex that decelerate and reverse its westerly winds; strong vortex events, which are characterized by anomalous strengthening of the vortex (Limpasuvan et al., 2005; Tripathi et al., 2015); and final warmings, which denote the breakdown of the polar vortex in spring or early summer until the subsequent autumn (Black et al., 2006). The observed surface response following such polar vortex events generally resembles the so-called Northern and

Southern Annular Modes (NAM/SAM; Thompson and Wallace, 2000; Baldwin and Dunkerton, 2001), or the North Atlantic



Oscillation (NAO; Ambaum and Hoskins, 2002; Charlton-Perez et al., 2018; Domeisen, 2019), which captures the tropospheric response specifically over the North Atlantic, where northern hemisphere (NH) stratospheric influence tends to be the strongest (Butler et al., 2017; Dai and Hitchcock, 2021). The surface temperature and precipitation responses associated with these large-scale tropospheric circulation patterns have been shown to contribute to extreme events such as cold air outbreaks and
precipitation extremes (Domeisen and Butler, 2020).

In the tropical stratosphere, the absorption of vertically propagating tropical waves from the troposphere below drive the alternating phases of easterly and westerly winds of the Quasi-Biennial Oscillation (QBO; Baldwin et al., 2001). In turn, the QBO has the ability to modulate the tropospheric jet streams, tropical convection, and other phenomena such as the Madden-Julian Oscillation (e.g., Gray et al., 2018; Kim et al., 2020; Anstey et al., 2021, and references therein). These tropospheric
teleconnections of the QBO are known to affect the NAO, as well as surface temperatures and precipitation near East Asia and in the tropics on both seasonal-mean and subseasonal timescales (Anstey et al., 2021; Gray et al., 2018; Haynes et al., 2021; Elsbury et al., 2021; Park et al., 2022). The QBO also has an apparent influence on the strength of the polar vortex, particularly in the Northern Hemisphere (NH), whereby easterly or westerly winds in the tropical lower stratosphere tend to lead to a weaker or stronger polar vortex, respectively (e.g., Holton and Tan, 1980; Calvo et al., 2007; Garfinkel et al., 2012;
Anstey and Shepherd, 2014; Gray et al., 2018; Rao et al., 2020a). This is known as the "Holton-Tan" effect; there are several mechanisms that help explain such QBO-polar vortex coupling, but generally they are tied to the QBO winds affecting the propagation and dissipation of waves in the polar stratosphere.

Since the stratosphere and its variations generally exhibit a higher persistence and predictability than the troposphere (Domeisen et al., 2020a; Son et al., 2020), their downward influence can lead to opportunities for long-range prediction of
surface weather on S2S timescales (Butler et al., 2019a). For example, all of the extreme polar vortex events mentioned above have shown potential for improved S2S surface prediction: SSWs (Sigmond et al., 2013), strong vortex events (Tripathi et al., 2015; Domeisen et al., 2020b), and final warmings (Butler et al., 2019b; Byrne et al., 2019). Tropical stratospheric variability associated with the QBO has also shown the potential to improve surface prediction on these time-scales (Marshall and Scaife, 2009; Garfinkel et al., 2018; Martin et al., 2021b). However, surface prediction skill is not always improved based on the con-
ditions in the stratosphere. In fact, several regions exhibit poorer predictability after extreme polar vortex events as compared to periods without (Domeisen et al., 2020b). There are likely several reasons for these shortcomings in dynamical predictions related to the inherent chaotic nature of dynamical coupling within the atmosphere. For example, not all SSWs appear to have a significant downward impact (Karpechko et al., 2017), likely due to tropospheric internal variability (Domeisen et al., 2020c; Afargan-Gerstman and Domeisen, 2020) and the duration and strength of anomalies in the lower stratosphere following the
SSW (Maycock and Hitchcock, 2015; Karpechko et al., 2017; Charlton-Perez et al., 2018; Domeisen, 2019; White et al., 2020).

Model biases and related shortcomings in simulating the stratosphere can affect dynamical predictions of stratospheric variability and stratosphere-troposphere coupling (Domeisen et al., 2020b; Charlton-Perez et al., 2013). Indeed, tropospheric biases identified independently of the stratosphere can sometimes be traced back to stratospheric anomalies in subseasonal forecasts; in the ECMWF extended-range prediction system, the persistence of the NAO was found to be constrained too strongly by the
state of the polar vortex at initialization (Kolstad et al., 2020). Since stratospheric variability and extreme events are primarily




governed by wave-mean flow interactions, biases in wave driving due to either resolved or parameterized processes can lead to biases in the mean state of the stratosphere, which can further alter its response to subsequent forcing (McLandress et al., 2012; Richter et al., 2014) and limit stratospheric predictability (Portal et al., 2021). More fundamentally, a model's vertical resolution in the stratosphere can influence its ability to realistically represent the stratosphere. For example, climate models

with limited resolution in the stratosphere tend to dramatically underestimate stratospheric variability (Charlton-Perez et al., 2013; Shaw et al., 2014; Richter et al., 2020; Rao et al., 2020a). Similarly, subseasonal forecast systems with severely limited model resolution in the stratosphere exhibit poor skill in predicting extreme polar vortex events beyond 1 week, whereas systems with higher model lids and higher vertical resolution in the stratosphere can predict these events up to 2-3 weeks in advance (Domeisen et al., 2020a). Subseasonal forecast systems that poorly represent tropospheric stationary waves also

tend to have low vertical resolution in the stratosphere, connected to larger stratospheric biases (Schwartz et al., 2021). On seasonal timescales, surface prediction skill can also in part be traced back to model properties such as vertical resolution in the stratosphere (Butler et al., 2016; Portal et al., 2021).

It is expected that identifying and properly addressing the sources of stratospheric biases would not only improve prediction in the stratosphere, but also help improve the surface prediction skill associated with the aforementioned stratospheric phe-

nomena. For example, some recent studies have investigated the impact of explicitly correcting stratospheric biases in model simulations, and found improvements to SSW statistics (Tyrrell et al., 2021) and the representation of the Holton-Tan effect (Karpechko et al., 2021). In other similar studies, teleconnections to the polar vortex related to Siberian snow cover (Tyrrell et al., 2020) and the El Niño-Southern Oscillation (ENSO; Tyrrell and Karpechko, 2021) were shown to be sensitive to model stratospheric biases. In the case of the Siberian snow cover, explicitly correcting the stratospheric biases had modest impacts

on the magnitude and duration of downward coupling between the polar vortex and the AO/NAO. However, in the case of ENSO teleconnections, correcting stratospheric biases had no detectable impact on the NAO, possibly because the NAO response to the strong ENSO events in the experiments was dominated by the tropospheric ENSO pathway to the North Atlantic (Jiménez-Esteve and Domeisen, 2018).

Given the role the stratosphere can play for skillfully predicting surface weather at subseasonal to seasonal timescales, it

is crucial to investigate and diagnose the stratosphere and stratosphere-troposphere coupling biases present in such models. Identifying and comparing these biases will serve as a data point for improving subseasonal forecast systems, and ultimately prediction skill, both in the stratosphere and for extended-range surface weather. Herein we perform a comprehensive investigation and intercomparison of stratosphere-related biases affecting subseasonal prediction systems. The present work is the result of a volunteer collaborative effort of the World Climate Research Programme (WCRP) Stratosphere-troposphere Pro-

cesses and their Role in Climate (SPARC) Stratospheric Network for the Assessment of Predictability (SNAP) activity, which is also the stratosphere sub-project of the WCRP/World Weather Research Programme S2S Prediction Project. In Section 2 we describe the datasets and methods we use to identify biases. We describe our results in Section 3, and discuss and summarize our findings in Section 4.





## 2 Data & Methods

### 2.1 Subseasonal-to-Seasonal (S2S) Hindcast and Reanalysis Datasets


We primarily use ensemble hindcast data from the S2S Prediction Project Database (Vitart et al., 2017). Where possible, we also include results from other ensemble forecast systems that do not provide data to the S2S database; these include the National Oceanic and Atmospheric Administration's Global Ensemble Forecast System version 12 (NOAA GEFSv12; Hamill et al., 2021; Guan et al., 2021), the Geophysical Fluid Dynamics Laboratory Seamless System for Prediction and

EArth System Research (GFDL-SPEAR; Delworth et al., 2020), the National Center for Atmospheric Research Community Earth System Model version 2 (CESM2) with version 6 of the Community Atmosphere Model as its atmospheric component (NCAR CESM2-CAM6, hereafter CESM2-CAM or CESM2-C), and CESM2 with the version 6 Whole Atmosphere Community Climate Model as its atmospheric component (CESM2-WACCM6, hereafter CESM2-WACCM or CESM2-W; Richter et al., 2021).

Table 1 lists the different S2S forecast systems, and includes relevant information about their hindcast and model configurations. Because the hindcast periods available for different systems vary substantially, unless noted otherwise, we limit our analyses to the 1999-2010 period, which is common to nearly all systems. The one exception is GEFSv12, which has hindcasts that span the period from 2000-2019, for which we simply use the 2000-2010 period. In order to balance including as many prediction systems as possible while also being inclusive of contributions from co-authors with differing capabilities to access

and store the large datasets involved, we designated the S2S database systems that provide at least 35-day forecasts as our "core systems", while others were considered optional. Thus, the number of S2S systems shown for a given analysis may vary, but will always include the seven core systems from the S2S database.

There are often multiple model versions of hindcast data available within the S2S database, representing updates to the forecast systems. These model updates sometimes include changes that can reasonably be expected to affect behavior in the

stratosphere, such as increases in the number of model levels or lid height, or changes to the atmospheric initialization. Table 1 lists the specific model versions we use; the important ones to note are for the CMA, ECMWF, and ECCC systems. For ECMWF we consider only CY43R1, CY43R3, and CY45R1; we exclude CY46R1 as it was the first cycle to use initialization from the ERA5 reanalysis. We also exclude cycles later than CY47R1 because these include updates that increase the order of vertical interpolation in the model that explicitly affect mean stratospheric biases (Polichtchouk et al., 2020). The CMA and

ECCC data both have model versions available in the S2S database that include changes to significantly higher model tops. Since we consider CMA to be a core system (as it performs forecasts beyond 35 days), we use the older low-top version known as the "BCC-CPS-S2Sv1" because the more recent "BCC-CPS-S2Sv2" hindcast period only covers 2004-2018. In contrast, ECCC data is considered to be optional, so we sometimes mix the use of the high- and low-top versions; in such cases, we explicitly describe these as "ECCC-hi" or "ECCC-lo", respectively (see Section 2.2 for the specific definitions of high versus

low top).

For the analyses herein, we use basic meteorological fields, including zonal and meridional wind components, temperatures, and geopotential heights. These fields from the S2S database are provided once daily at instantaneous 00 UT verification



**Table 1.** Details of the subseasonal-to-seasonal forecast systems used herein.

| Model | S2S Database Version(s) | Hindcast Period | Ensembles | Forecast Span | Model Top |
|---|---|---|---|---|---|
| BoM*† | POAMA P24 | 1981-2013 | 33 | 62 days | 10 hPa |
| CESM2-CAM* | – | 1999-2019 | 11 | 45 days | 2 hPa |
| CESM2-WACCM | – | 1999-2019 | 5 | 45 days | $4.5 \times 10^{-6}$ hPa |
| CMA*† | BCC-CPS-S2Sv1 | 1994-2014 | 4 | 60 days | 0.5 hPa |
| CNR-ISAC* | GLOBO | 1981-2010 | 5 | 32 days | 6.8 hPa |
| CNRM† | CNRM-CM 6.0 | 1993-2014 | 15 | 47 days | 0.01 hPa |
| ECCC-hi | GEPS 6 | 1998-2017 | 4 | 32 days | 0.1 hPa |
| ECCC-lo* | GEPS 4 | 1995-2014 | 4 | 32 days | 2 hPa |
| ECMWF† | CY43R1, CY43R3, and CY45R1 | 1996-2018 | 11 | 46 days | 0.01 hPa |
| GEFSv12 | – | 2000-2019 | 11 | 35 days | 0.1 hPa |
| GFDL-SPEAR* | – | 1991-2020 | 15 | 12 months | 1 hPa |
| JMA | GEPS1701 | 1981-2012 | 5 | 34 days | 0.01 hPa |
| KMA† | GloSea5-GC2 | 1991-2010 | 3 | 60 days | 85 km |
| NCEP† | CFSv2 | 1999-2010 | 4 | 44 days | 0.02 hPa |
| UKMO† | GloSea5 and GloSea5-GC2-LI | 1993-2016 | 7 | 60 days | 85 km |

CESM2-WACCM hindcast initializations were only performed for September-April

† Core systems

* Systems with low-top models

times on a 1.5° × 1.5 ° latitude/longitude grid, with 10 pressure levels between 1000 and 10 hPa (with about 3 levels in the stratosphere, including 100, 50, and 10 hPa). GEFSv12 fields are provided 6-hourly on a 0.5° × 0.5 ° grid, with 25 pressure
levels between 1000 and 1 hPa (6 between 100 and 10 hPa). GFDL-SPEAR fields are also 6-hourly on a 0.5° × 0.5 ° grid, with 33 pressure levels between 1000 and 1 hPa. For the CESM2-CAM and CESM2-WACCM data, we make use of the available "zonal mean" collection of variables that are provided as daily means on pressure levels closest to the model levels; we interpolate this data to a set of 32 standard pressure levels between 1000 and 10 hPa (with 6 between 100 and 10 hPa); these fields have 192 latitudes (~0.9424° resolution) from the finite volume grid.

The subseasonal hindcast datasets used are all initialized with different atmospheric analyses. Therefore, to ensure that comparisons and biases are all determined with respect to a consistent dataset, we compare hindcast fields to those from the ERA-Interim (ERA-I) reanalysis (Dee et al., 2011). Different modern reanalysis products generally agree very well amongst one another for the time periods (post-1999) and levels (10 hPa and below) we consider (Long et al., 2017; Gerber and Martineau, 2018; Fujiwara, M. et al., 2021), and thus our results are unlikely to be sensitive to the choice of reanalysis.





## 2.2 Methods

Throughout the paper we distinguish between forecast systems with low- and high-top atmospheric models. We define the systems having model tops at or above 0.1 hPa with several levels above 1 hPa as high-top; any others that do not meet these criteria are specified as low-top. In total, we consider 8 high-top and 6 low-top models (see Table 1), though the precise number of models included in a given analysis varies. In our figures, we highlight low-top models with asterisks and/or dashed lines.

For a given diagnostic, raw biases among the forecast systems are computed by taking the difference between the ensemble-mean hindcasts and ERA-I. We generally composite these biases according to lead time and/or season in order to determine the systematic differences of the hindcast predictions from reanalysis. For some applications we derive lead-time dependent climatologies for the different forecast systems, which we use to determine forecast anomalies and to apply bias-correction. For systems in the S2S database, these climatologies are found by averaging all ensemble-mean hindcasts for a given day of year at a specific lead time. For systems that provide a fixed set of hindcast initializations that do not uniformly cover the same days of year in the hindcasts (such as the GEFSv12 and the CESM2 systems), we create the lead-dependent climatologies according to the method outlined in Pegion et al. (2019); briefly, this method involves averaging all hindcasts for a given day of year and lead time (which is generally less than the total number of years in the hindcast archive), and then applying a rolling 31-day average with centered triangular weights to the "raw" and noisy lead-dependent climatology. Hindcast anomalies are then determined by subtracting these climatologies for a given day of year and lead time from the raw forecast quantities. In some cases we apply a linear bias-correction to raw quantities in order to remove any climatological drift that may exist. This is done by removing the difference between each system's lead time-dependent climatology and the ERA-I climatology from the predicted quantity in question: $Q_{BC}(t,l) = Q_{Raw}(t,l) - [\bar{Q}_{hc}(t_{doy},l) - \bar{Q}_{obs}(t_{doy}+l)]$, where $Q_{BC}$ is the bias corrected quantity, $Q_{Raw}$ is the raw quantity, $\bar{Q}_{hc}$ is the hindcast climatology, $\bar{Q}_{obs}$ is the observed/reanalysis climatology, $t$ is the forecast initialization date, $l$ is the lead time, and $t_{doy}$ is the day of year of the initialization date $t$.

In sections 3.3 we investigate whether there are biases in relation to extreme stratospheric events, including SSWs and strong vortex events in the NH. SSWs are defined using ERA-I data, based on reversals in the 10 hPa 60°N zonal mean zonal winds, consistent with Charlton and Polvani (2007) and Butler et al. (2017). Similarly to SSWs, strong vortex events are defined using ERA-I data when the daily mean 10 hPa 60°N zonal mean zonal winds exceed 41.2 m/s for at least two consecutive days, corresponding to approximately the 80th percentile of the November-March 1980-2012 zonal winds, consistent with the definition used by Tripathi et al. (2015). These definitions have also been employed in prior investigations of the stratosphere in S2S systems (Domeisen et al., 2020a, b). The central dates of these events are considered to be the first day on which these zonal wind thresholds are met.



## 3 Results

### 3.1 Global and Zonal Mean Biases

From the perspective of model evaluation, examining mean state biases in the stratosphere is useful since the processes that govern the distribution of zonal mean temperatures and winds in the stratosphere are quite well understood. Global- and annual-mean temperatures in the stratosphere should be very close to radiative equilibrium (e.g., Olaguer et al., 1992). In contrast, seasonal and meridional variations in zonal mean stratospheric temperatures and winds arise from dynamic influences such as wave-mean flow interaction. Meridional circulations driven by the dissipation of waves drive stratospheric temperatures away from radiative equilibrium, which affects the zonal-mean zonal winds through thermal-wind balance (e.g., Andrews et al., 1987; Shepherd, 2000). Model biases in global annual-mean stratospheric temperatures, and/or seasonally-varying biases in zonal mean temperatures and winds therefore provide information about the likely origin of model errors, either pointing to model components that affect radiative processes, or those that affect middle atmosphere dynamical processes such as parameterized gravity wave drag.

We first consider the annual, global mean temperature biases among the S2S forecast systems. Figure 1 shows these biases for 10, 50, and 100 hPa for each of the models at a lead time of 4 weeks (days 22-28). These biases generally develop and increase in magnitude monotonically with lead time, with many of these biases being present at earlier lead times of 1-2 weeks (not shown). The magnitude of biases tend to be largest in the middle stratosphere at 10 hPa, with 6 of the models shown exceeding absolute biases over 2 K. Positive temperature biases tend to be most common across the models and levels, with the exception of ECMWF and CESM2-WACCM, which primarily have negative temperature biases.

There are some apparent differences between the annual, global mean stratospheric temperature biases between the systems with high- versus low-top models (denoted with asterisks). Warm biases are more common across the pressure levels in the low-top systems, and the highest magnitude biases are generally at 10 hPa, which is likely related to the model tops being relatively close to this level. Some of the low-top systems (CMA-S2Sv1, and CNR-ISAC) have biases that are much less severe in the lower stratosphere and more comparable to the high-top systems. The biases for the high-top systems are generally smaller in magnitude, but there are some exceptions: for instance, the CESM2-WACCM, ECMWF, GEFSv12, and NCEP systems have biases at some levels that are as large or larger in magnitude than the low-top systems. Figure 1 also highlights some "familial" relationships . In particular, the "NOAA-family" of forecast systems (including GEFSv12, GFDL-SPEAR, and NCEP) all show global mean warm biases throughout the lower and middle stratosphere. The biases in the KMA and UKMO systems are very similar, likely owing to the fact they both use the same GloSea-5 model (see Table 1). There are large differences between the biases apparent in CESM2-CAM and CESM2-WACCM, which may be due to a combination of factors: aside from the differences in model tops, CESM2-WACCM includes fully interactive tropospheric and stratospheric chemistry (Gettelman et al., 2019; Richter et al., 2021), and the atmospheric models are initialized with different reanalyses ("CFSv2" for CESM2-CAM, and "MERRA-2" for CESM2-WACCM).

The zonal mean biases across the models are broadly consistent with the annual global mean temperature biases shown above, but reveal further important details about their vertical and meridional structures. Figure 2 shows the zonal mean



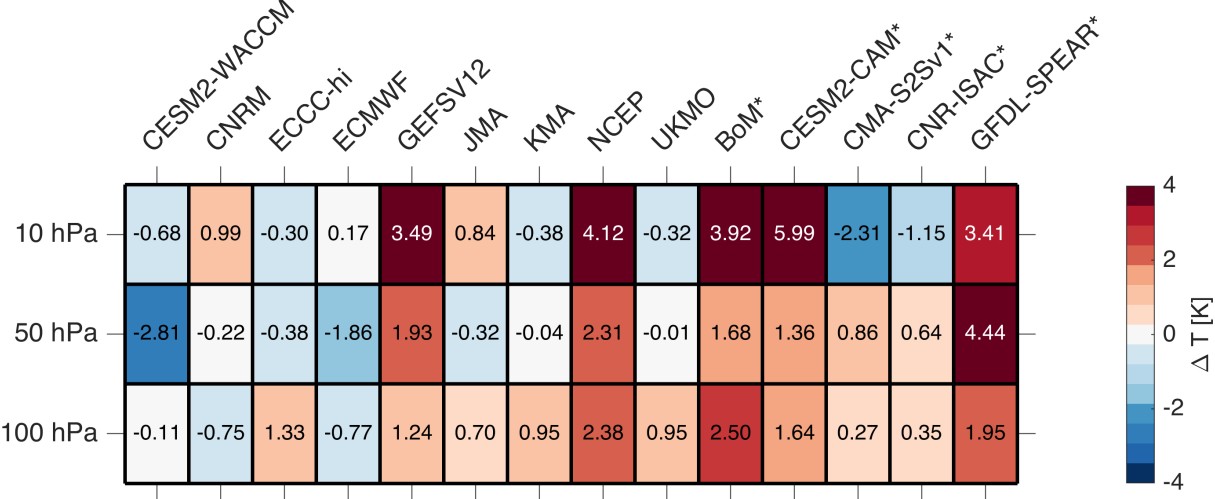

**Figure 1.** Global- and annual-mean temperature biases at a lead time of 4 weeks. The columns show individual forecast systems, while the rows correspond to different pressure levels. The numerical values displayed are shown in Kelvin. Asterisks at the end of names denote the systems with low-top models.

biases in temperatures and winds as a function of latitude and height for different seasons, composited for high- and low-top models at a lead time of 4 weeks. These biases are shown for individual models in the Supplemental figures. We note that

the interpretation of vertical variations in Figure 2 and the supplemental figures requires some caution since the data from the S2S database are only provided on roughly 3 levels in the stratosphere (100, 50, and 10 hPa). Furthermore, since the UKMO and KMA systems both make use of the GloSea-5 model with the same atmospheric configurations, KMA has been left out of the high top composite in Figure 2 so as not to unfairly weight the high-top composite; as the supplemental Figures 13 and 15 show, these systems have nearly identical biases. The patterns of biases in temperatures and zonal winds are consistent

across the high- and low-top composites with signatures of (1) global mean warm biases in the stratosphere (consistent with Figure 1), (2) cold extratropical UTLS biases in both hemispheres, (3) easterly wind biases in the tropical stratosphere, and (4) too strong/cold stratospheric polar vortices in the winter hemispheres. It is clear, however, that the biases in the low-top models are generally much larger in magnitude despite having similar patterns.

The cold extratropical UTLS and cold winter pole/strong polar vortex biases are recognized as longstanding issues in forecast

and climate models. The former issue is generally thought to be related to excessive longwave cooling from a moist bias that is present in initial conditions, and/or develops over time due to an inability to properly maintain the distribution of water vapor in the region of the tropopause (see, e.g., Bland et al., 2021, and references therein). Figure 2 and the supplemental figures show that these cold biases are still apparent in high-top models, but are generally smaller in magnitude, particularly in the winter hemispheres. This may be related to the fact that the systems with high-top models generally have more model

levels/higher vertical resolution, making them better able to represent processes in the tropopause region. The too strong/cold winter stratospheric polar vortex biases, particularly in the NH, reflect a lack of dynamical variability. Figure 2 shows that the



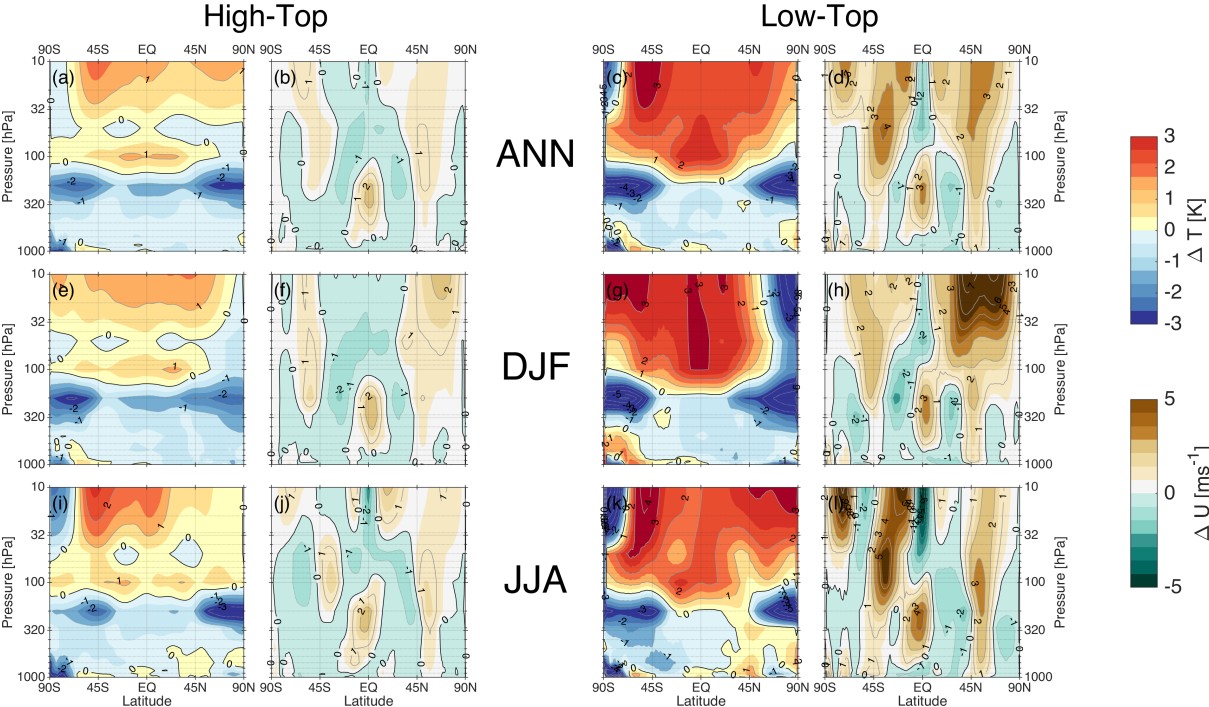

**Figure 2.** Zonal mean (a,c,e,g,i,k) temperature (b,d,f,h,j,l) zonal wind biases as a function of (vertical axes) pressure and (horizontal axes) latitude at a lead time of 4 weeks. Data are averaged over (a-d) the full annual cycle, (e-h) boreal winter (DJF) and (f-l) austral winter (JJA), and are shown separately for (a,b,e,f,i,j) high-top and (c,d,g,h,k,l) low-top models.

low-top systems have much more pronounced cold winter poles and stronger polar vortex winds in both hemispheres, which is consistent with prior studies that find models with tops below the stratopause generally fail to realistically simulate stratospheric variability (e.g., Charlton-Perez et al., 2013; Shaw et al., 2014; Rao and Garfinkel, 2021a). There are multiple reasons why

these biases are generally worse in low-top models, usually related to an underestimation of resolved/parameterized gravity wave drag, as well as possible unphysical effects of the model lid (e.g., Shaw and Perlwitz, 2010; Richter et al., 2014).

The apparent easterly bias in the tropical stratospheric zonal winds across the high- and low-top systems represents a general bias among the models in representing the QBO. We examine these biases in the next section in more detail.

### 3.2 Biases Related to the Tropical Stratosphere and QBO

The circulation of the tropical stratosphere is dominated by the QBO, and hence the zonal mean biases in the tropics shown above must be partially related to each forecast system's ability to maintain the QBO from initial conditions. To the extent that S2S models can represent the QBO and its teleconnections accurately, the QBO could lead to more reliable predictions on subseasonal and seasonal timescales in the troposphere (Garfinkel et al., 2018; Merryfield et al., 2020). Therefore, we now consider whether this potential is realized by the S2S forecast systems. Unlike elsewhere in the paper, here our QBO





analyses make use of the full hindcast periods available to each model to maximize the number of QBO cycles available when
determining biases (see Table 1).

Figure 3 considers whether these models are capable of maintaining the anomalous QBO winds present in initializations from
November through February (NDJF). In the top row, we composite hindcasts in which the zonal wind anomalies (computed
with respect to each model's climatology) at 10 hPa, 5°S-5°N averaged over the first three days of the hindcast exceed 3 m/s as
being westerly QBO (WQBO), and less than -3 m/s as easterly QBO (EQBO), with ERA-I reanalysis data subsampled to match
the dates included in each model's composite. The middle row shows similar quantities, but with the composites determined
using the zonal wind anomalies at 50 hPa instead. At 10 hPa (Fig 3, top row), the systems with low-top models (dashed lines)
clearly struggle to maintain the amplitude of both EQBO and WQBO initialized anomalies, which decay toward 0 at longer
lead times. The systems with high-top models generally show much slower and slighter decay, with anomalies that more closely
match reanalysis across lead times. Results are similar but more pronounced at 50 hPa (middle row). The decay of anomalies at
50 hPa are particularly apparent in the EQBO composite, for which the wind anomalies in most systems decay by nearly 5 m/s
or more toward climatology by the end of the forecasts. The main exceptions are the UKMO and KMA systems, which share
the same atmospheric models; these both perform well relative to others, as demonstrated by their anomalies at different lead
times closely matching reanalysis. Overall, the fact that the subseasonal forecast systems are generally better able to simulate
and maintain the QBO in the middle stratosphere versus the lower stratosphere is similar to what has been found in climate
models, seasonal forecast systems, and models participating in the "QBO-initiative" (QBOi; Richter et al., 2020; Rao et al.,
2020a; Bushell et al., 2020; Stockdale et al., 2020). This difference in success between the mid- and lower-stratospheric QBO
could be related to a couple of factors, including the need for high vertical resolution in the lower stratosphere for realistically
capturing upward wave flux and subsequent downward QBO propagation, as well as possible influences from model vertical
diffusion in the lower stratosphere (Geller et al., 2016; Garfinkel et al., 2021; Polichtchouk et al., 2021).

The QBO has been shown to influence tropical convection on subseasonal timescales, and one of the leading mechanisms for
this effect is related to the QBO's mean meridional circulation, which leads to temperature (and buoyancy frequency) anomalies
in the tropical tropopause layer that subsequently affect high clouds and convection (Gray et al., 2018). These relationships
also underpin the QBO's observed relationship with the Madden-Julian Oscillation (MJO; Yoo and Son, 2016; Son et al., 2017;
Lee and Klingaman, 2018; Martin et al., 2021a), which can modulate MJO forecast skill (Kim et al., 2019; Lim et al., 2019). To
this end, we examine the associated temperature anomalies in the lowermost tropical stratosphere (i.e., at 100 hPa; bottom row
of Figure 3). These biases are composited based on the initial QBO wind anomalies at 50 hPa (consistent with the middle row).
The temperature anomalies are proportional to the QBO-related shear in the lower stratosphere with typical WQBO/EQBO
wind anomalies corresponding to warm/cold anomalies of ±0.5 K. These temperature anomalies are present in the initialized
states of the models, but in most they decay with leadtime. The rate of weakening differs among the models, with the low-top
systems generally showing much more rapid decay of anomalies. There appears to be little difference between the EQBO and
WQBO composites in the ability of models to maintain the tropical lower stratosphere temperature anomalies. Underestimating
the amplitude of the QBO in the lower stratosphere (both in winds and temperatures) down to the tropopause is an issue similar

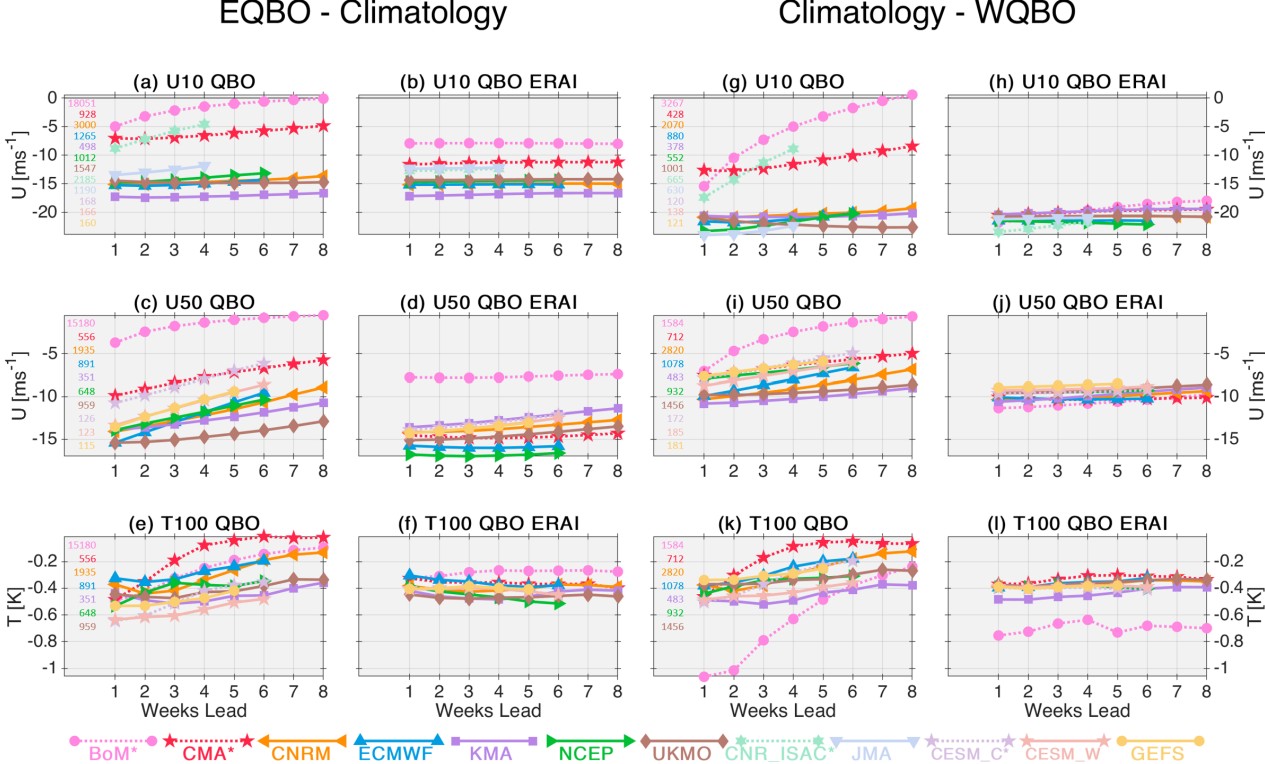

**Figure 3.** Hindcast composite time series of tropical stratospheric anomalies in the easterly and westerly phases of the QBO at lead times from 1 to 8 weeks (a,c,e,g,i,k), and corresponding ERA-I data subsampled to match the days covered by each hindcast ensemble (b,d,f,h,j,l). (top row) 10 hPa, 5°S-5°N zonal wind; (middle row) 50 hPa, 5°S-5°N zonal wind; (bottom row) 100 hPa, 5°S-5°N temperature. All composites are based on November-February initializations only, with the number of initializations in each composite shown in the first and third columns.

to that present in CMIP6 and QBOi models (Bushell et al., 2020; Richter et al., 2020), and may be a contributing factor to why

many subseasonal forecast systems show insignificant relationships between the QBO and MJO (e.g., Kim et al., 2019).

The QBO is known to have an important teleconnection to the boreal winter polar vortex known as the "Holton-Tan effect" (e.g., Holton and Tan, 1980; Baldwin et al., 2001; Calvo et al., 2007; Garfinkel et al., 2012; Anstey and Shepherd, 2014), in which weaker or stronger polar vortex winds preferentially occur with EQBO or WQBO winds in the lower stratosphere, respectively. Figure 4 shows the composite anomalies in 10 hPa polar vortex winds, and 100 hPa polar cap geopotential heights

(Z100) composited based on the 50 hPa QBO phase at initialization. Since the Holton-Tan effect develops in early winter and is most pronounced in mid-winter, we limit our focus to forecast initializations within November and December, before the effect is well-established in the boreal winter. In the reanalysis EQBO composites, the vortex weakening signal is apparent (with some differences being due to sampling from different model initializations), with winds that weaken by 5 to 10 m/s beyond 3 weeks (Fig 4b). However, it is clear that the subseasonal forecast systems underestimate the magnitude of polar



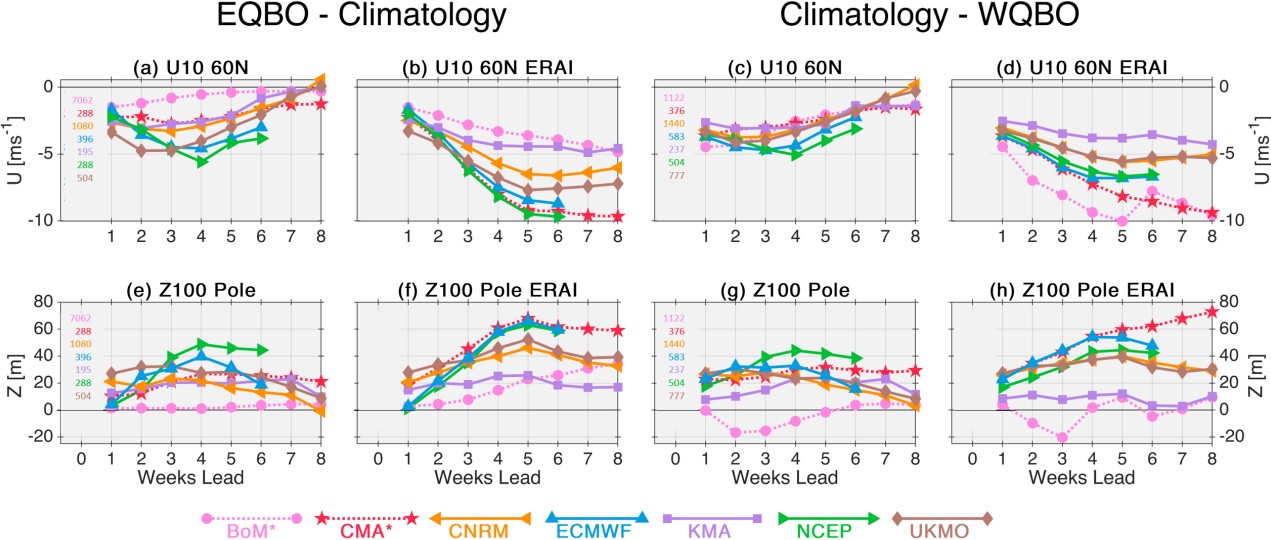

**Figure 4.** As in Figure 3, but with QBO composites based on the initial 50 hPa winds from November and December initializations. (top row) zonal wind at 10h Pa, 60°N; (bottom row) 100hPa, 70°N-pole polar cap geopotential heights (Z100). The number of initializations in the composites are shown in the first and third columns.

vortex weakening (Fig 4a). In some systems, the weak vortex anomaly grows stronger out to around week 3 (see, e.g., NCEP, UKMO, and ECMWF), but all the anomalies decay thereafter. The results for WQBO are largely consistent, but note that the differences are expressed as climatology minus WQBO, and hence the anomalies have a negative sign. The bottom row of Figure 4 shows the anomalies in Z100, where persistent anomalies in the strength of the lower stratospheric vortex are more closely tied to surface-related impacts. Here the results are similar to the 10 hPa polar vortex winds; while most models are initialized with higher/lower Z100 in EQBO/WQBO, they generally fail to simulate the change in amplitude of the anomalies with lead time.

### 3.3 Northern Hemisphere Polar Vortex Variability

Variability in the NH polar winter stratosphere primarily arises from extreme dynamical polar vortex events, including midwinter SSWs and strong vortex events. The occurrence of these events are generally associated with extremes in upward wave driving on the polar vortex, with SSWs/strong vortex events being preceded by extended periods of above/below-normal wave driving, respectively (e.g., Polvani and Waugh, 2004). A typical metric for upward planetary wave driving is the meridional eddy heat flux, $v'T'$ (with $v$ being the meridional wind, $T$ the temperature, and the primes denoting deviations from the zonal mean). In models, such wave driving should be resolved, dependent upon tropospheric variability and proper simulation of vertical wave propagation. There can also be significant variability among polar vortex events in terms of different characteristics, such as their timing, magnitude, persistence, and even polar vortex geometry (e.g., Karpechko et al., 2017). Importantly, the





occurrence of such polar vortex events can lead to long-term coupling with the troposphere; in forecast models, the occurrence of these events can improve tropospheric predictability by providing "forecast windows of opportunity" (e.g., Butler et al., 2019a, and references therein).

We first examine and compare biases in the NH eddy heat fluxes among the subseasonal forecast systems. Figure 5 shows
the December-February eddy heat flux biases with respect to ERA-I at a lead time of 4 weeks. The climatological DJF 100 hPa heat flux from ERA-I is shown in Figure 5a, while the high- and low-top composite biases are shown in b and c. The observed climatological heat fluxes show two centers of action, with one over the North Pacific, and another over Scandinavia/Siberia. The mean biases strongly differ between the high- versus low-top composites. The high-top systems underestimate heat fluxes in the Pacific region more than over Scandinavia/Siberia, but none of the biases are statistically significant. In contrast, the
low-top systems significantly underestimate heat fluxes in both regions, while also overestimating heat fluxes over Canada and Greenland; the latter likely indicates the low-top systems do not capture the negative heat fluxes seen in the ERA-I climatology. The representation of heat fluxes in the Scandinavian/Siberian region has been shown in previous work using S2S models to be deficient (Garfinkel et al., 2019; Lee et al., 2020), and thus we show the evolution of the biases in this region in Figure 5d. In most models the week-1 Scandinavian/Siberian heat flux biases are small in magnitude, especially for the high-top
models. However, by weeks 3-4, some of the models have well-developed biases that stay relatively constant, at least out to week 6. The low-top systems show large magnitude negative biases that indicate they underestimate the Scandinavian/Siberian heat fluxes across different lead times. Here the NCEP system stands out among the rest, since it actually overestimates the Scandinavian/Siberian heat flux in weeks 4-6. The results are similar for the heat flux biases over the Pacific sector in Figure 5e; overall, the ECMWF and CNRM systems appear to be the only high-top systems that consistently underestimate the heat fluxes
in both the Pacific and Scandinavian/Siberian regions, albeit to a much lesser extent than the low-top systems. In Figure 5f we show biases in the heat fluxes due to zonal waves-1 and 2, since these planetary scale waves contribute most to the total eddy heat flux. Here, the high-top models generally have small biases near 0, indicating that on a global scale, the regional biases tend to cancel. This is not the case for the low-top models, which still show a consistent underestimation of heat fluxes across different lead times, which is consistent with the strong vortex/cold pole bias shown in the low-top zonal mean composite from
Figure 2.

The biases in the stratospheric background circulation (section 3.1) and heat fluxes described above can affect the occurrence and timing of threshold events such as SSWs and strong vortex events. Figure 6 shows the probability of occurrence of 10 hPa 60°N winds less than 0 m/s (corresponding to SSWs) or greater than 41.2 m/s (corresponding to strong vortex events) for different weekly lead times. These probabilities are composited based on initialization, so for instance, the week 4 values of
the January initializations (Figure 6e,f) include verification times from the month of February. The seasonal distribution from reanalysis (circles) indicates a low probability of SSW events in early winter (November and December), with the highest occurrence of events being in late winter (January-March). Note that since the figure is composited based on initializations, the probabilities for March include verification times in April, and so the corresponding probabilities for easterly winds are likely influenced by final warmings. Regardless, this seasonal cycle is only partially reproduced in the subseasonal models,
which particularly underestimate the probability of events for December and January initializations at weeks 3 and 4, and



**Figure 5.** (a) The December-February climatology of ERA-I eddy heat fluxes, $v'T'$, at 100 hPa. (b) The high-top composite of eddy heat flux model biases with respect to ERA-I from November-January initializations at a lead time of 4 weeks. (c) is as in (b), but composited for the low-top models. In panel a, the line and colour-filled contours match the colorbar spacing of 10 ms$^{-1}$K; in panels b and c, the line contours match the colorbar contour intervals of 5 ms$^{-1}$K, but colours are only shown where the biases are statistically significant at the 95% level from a two-tailed Student's t-test. (d) Time series of the difference between the S2S hindcasts and ERA-I for $v'T'$ averaged over the Scandinavian/Siberian region (blue box in panels a-c), at lead times from 1 to 8 weeks. (e) is as in (d), but for the Pacific region (red box in panels a-c). (f) is also as in (d), but for the combined wave-1 and 2 planetary wave heat flux averaged over 45°-75°N

overestimate the probability for March initializations at weeks 3 and 4. This bias is in agreement with results from climate models (Ayarzagüena et al., 2020; Tyrrell et al., 2021), which tend to exhibit a peak in SSW occurrence in late winter instead





of in January, and seasonal prediction models, which also fail to reproduce the SSW peak in January (Portal et al., 2021) despite the seasonal average of SSWs often being well reproduced (Domeisen et al., 2015). Interestingly, the NCEP system

consistently predicts a higher occurrence of easterly winds than other systems. This means the NCEP system is more accurate for week 4 SSW risk forecasts initialized in December and January, but then overestimates SSW probabilities in February and March. The NCEP system's higher prediction of easterly winds is possibly related to its significant weak vortex bias (see Supplement Figure 14), which may be linked to its overestimation of eddy heat fluxes from the Scandinavian/Siberian region (Figure 5d).

Strong vortex events (right column of Figure 6) exhibit an entirely different seasonal cycle, with most events occurring between December and January (primarily due to the threshold-based definition and the climatological maximum strength of the vortex occurring in these months). Most models tend to underestimate the frequency of strong polar vortex winds for November and December initializations, particularly at weeks 3 and 4. There are some notable exceptions, including the CNR-ISAC, CNRM, GEFSv12, and CESM2-CAM systems, which all have substantial strong polar vortex biases in their wintertime

zonal winds (Supplemental Figures 2, 5, 6, and 10). For January initializations, most models instead overestimate the frequency of strong vortex winds beyond week 1 (Fig. 6f), which is also consistent with the general strong vortex biases evident in the model composites of Figure 2.

In addition to the probability of the events from raw forecasts, the black horizontal lines in Figure 6 indicate the probabilities estimated from bias-corrected forecasts (see Section 2.2 for details of the mean bias-correction process). The probability

of both SSWs and strong vortex events in the bias-corrected hindcasts initialised in November, December and January is generally either close to, or smaller than, the observed probabilities from ERA-I across all forecast systems. In most cases, this corresponds to an improvement over the raw forecasts. Especially for the prediction of SSWs for forecasts initialized in January (Fig. 6e), the mean bias correction clearly improves the estimates over those from the raw data, particularly for lead times of 3-4 weeks. However, the bias correction for late winter/early spring predictions (initializations in February and March, weeks

3 and 4) does not necessarily bring the easterly wind probabilities closer to observations. In some cases the bias correction increases the probability of events, even for systems whose un-corrected probabilities already closely match reanalysis. This may mean that model zonal-mean zonal wind biases in late winter and early spring tend to not dynamically alter the probability of zonal wind reversals at times when final warmings may be expected to occur. The exceptions here are the systems with the most severe biases, such as BoM and CNR-ISAC. Nevertheless, and especially for early winter, the magnitude of zonal wind

biases clearly changes the probability of fixed threshold events in most of the S2S systems. This supports the utility of bias correction for stratospheric S2S forecasts, albeit with some limitations. Furthermore, such bias correction has to be applied and interpreted with care, since the non-linear dynamics in the models evolve according to their own potentially biased mean states, and therefore, little can be said about potential tropospheric responses to such bias-corrected stratospheric forecasts.

There also exist biases in the magnitude of predicted events, even at relatively short lead times. In Figure 6 we identified the

probability of polar vortex events occurring in the subseasonal forecasts at different lead times; in Figure 7 we instead focus on observed polar vortex events in the ERA-I record, and assess the forecasted wind changes at verification times surrounding the observed events. Figure 7 shows the distributions of simulated wind changes associated with SSWs and strong vortex events



**Figure 6.** Probability of boreal (left column) sudden stratospheric warmings and (right column) strong vortex events, shown individually for composites of initializations within each month (rows) and weekly lead time (horizontal axis, alternating grey/white background shading within panels). For each model, the solid bars show the raw estimates, while the bold black horizontal lines indicate the probability determined after bias-correction. The coloured circles indicate the probabilities computed using ERA-I subsampled to match the same dates from each individual set of model hindcasts.

(defined as in Section 2.2) in the 1999-2010 reanalysis record. The deceleration or acceleration associated with these events is measured by computing the change in the hindcast/reanalysis zonal-mean zonal wind at 10 hPa and 60°N, at ±5 days around the ERA-I event onset dates. Almost all systems underestimate the wind changes for both SSW and strong vortex events at





all lead times, yielding dominantly positive/negative mean errors for SSWs/strong vortex events, respectively. However, the prediction of the observed magnitude of events clearly improves with decreasing lead time, as expected. At weeks 3 and 4 lead times, the predicted wind change distributions among the models are generally close to zero and exhibit small spread. This indicates that the prediction systems are generally not forecasting extreme vortex events at verification times around the

target event dates; instead they predict climatological zonal-mean zonal wind values or only weak wind tendencies of the same sign as the events. At shorter lead times within the typical deterministic predictability limit of roughly two weeks for the stratosphere, some systems still underestimate the magnitude and spread of the observed wind changes; many of these are the low-top models such as BoM, CESM2-CAM, and CMA. For SSWs, the ECMWF and NCEP systems have the smallest errors of around 5 m/s; for strong vortex events, the CNRM, CESM2-CAM, and GEFSv12 systems consistently have the lowest

mean errors within 10 m/s from 1-3 weeks, but these systems also have substantial positive zonal wind biases. Figure 7c,d also highlights the disparity in the magnitude of wind changes between SSWs and strong vortex events; median errors for SSWs are on the order of 15 - 20 m/s with outliers up to 60 m/s, whereas the range is much smaller for strong vortex events. This reflects the large and sudden deceleration of winds that occur during SSWs that (in absolute terms) is much larger than the acceleration of winds for strong events.

Finally, we examine whether there are biases among the subseasonal models in the geometry of the polar vortex for forecasts surrounding observed SSW events. Such characteristics are important since biases in the shape or location of the vortex would affect the vertical wave propagation that may ultimately influence the occurrence and/or magnitude of SSWs. We examine these characteristics using elliptical diagnostics, which provide quantities such as the vortex centroid latitude and aspect ratio (e.g., Waugh, 1997; Seviour et al., 2013) that can be used to quantify the displacement or stretch of the vortex during SSWs.

We perform these calculations using the hindcast 10 hPa geopotential height, assuming that the 30 km contour is representative of the vortex edge. Figure 8 shows the ensemble-mean predictions of the centroid latitude and aspect ratio diagnostics as a function of lead time and initialization with respect to composites of the central dates of displacement SSWs and split SSWs, respectively, within 1999-2010. Note that the colorbars of Figure 8 transition to pink colors at 66°N for centroid latitude, and 2.4 for aspect ratio, corresponding to the thresholds used in Seviour et al. (2013) to define displacement and split SSWs. For

displacement events (Fig 8a-i), most models capture a latitudinal deviation from the pole at long lead times of around a month, though with an underestimation of the magnitude of the displacement at lead times beyond 3 weeks. However, the BoM, CMA, and NCEP systems show signs of systematic biases in their predicted centroid latitudes. BoM and CMA (both low-top systems) show virtually no centroid latitude variability at longer lead times, with only the forecasts falling within about two weeks of the SSWs showing significant latitudinal displacements. On the other hand, NCEP appears to have a systematic bias toward

a vortex that is too frequently displaced at longer lead times. The results for these three models are consistent with our prior results that showed the BoM and CMA systems struggle to represent stratospheric variability (Figures 5,6, and 7), and that NCEP has a weak vortex bias and over-predicts the occurrence of SSWs (Figure 6 and supplemental Figure 14).

    In the case of vortex split SSWs (Figure 8 j-r), the high-top models perform much better relative to the low-top models, though still worse relative to displacement events. Vortex split events are known to be inherently less predictable than displace-

ment events (e.g., Taguchi, 2018; Domeisen et al., 2020a), but the low-top models only show enhanced aspect ratios within a



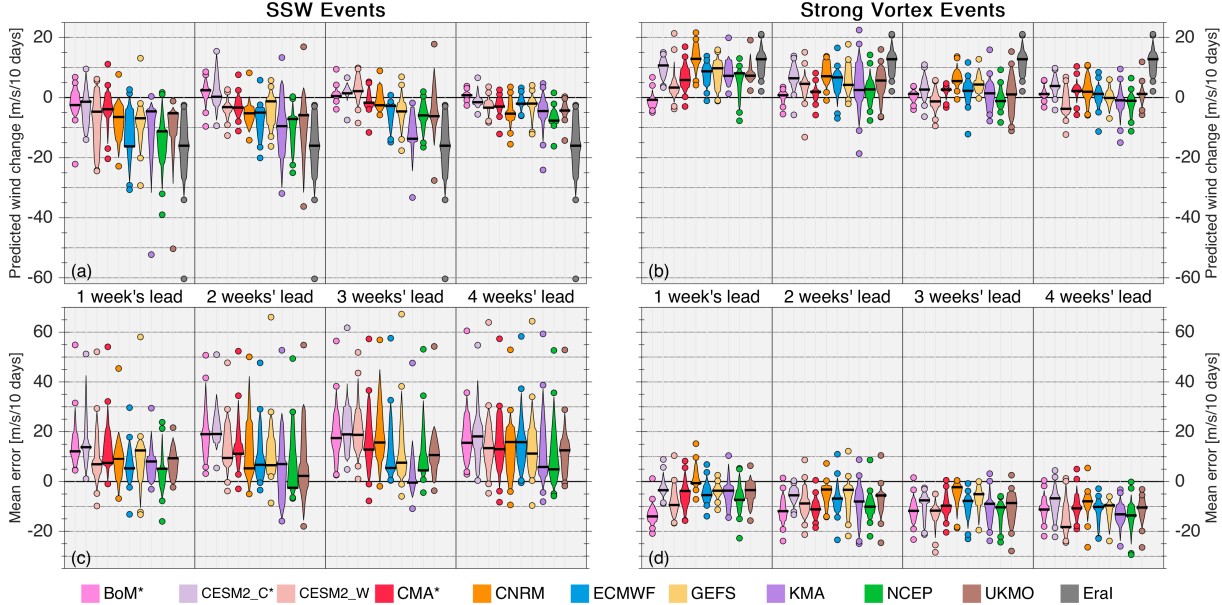

**Figure 7.** Distributions of the composited wind changes forecasted by each ensemble for forecast verification times surrounding (left column) boreal sudden stratospheric warmings and (right column) strong vortex events, composited over all such events in the 1999-2010 hindcast records. This corresponds to 11 SSW events and 13 strong vortex events across the boreal winters from 1999/2000 to 2009/2010. Units are m/s over the 10-day period centered on the observed event central dates within the forecasts or reanalysis. Within each section of each panel, data are shown as violin plots covering the 15-85% range of wind changes or mean errors, with outliers outside this range indicated with individual coloured circles, and black horizontal lines indicating the median value. In panels (a) and (b) the quantities shown are raw values, while those in panels (c) and (d) are shown as deviations from ERA-I. Note that individual models may contain a slightly different number of samples, especially for longer lead times, due to the different hindcast lengths available for each system (see Table 1); in other words, some hindcasts do not fully cover the 10-day periods surrounding the observed SSWs and strong vortex events, and are excluded.

lead time of roughly 10 days. Of these, the BoM system shows large aspect ratios only in the initializations that are close to the onset dates. While these diagnostics do show evidence of systematic biases among some of the modeling systems, the sample sizes under consideration here are quite small, with only 6 displacement and 3 split events in the common time period that is analyzed here.

## 3.4 Southern Hemisphere Polar Vortex Variability

In the Southern Hemisphere (SH), stratospheric polar vortex variability is mainly associated with interannual variability in the timing of the springtime polar vortex break-down (i.e., the final warming) sometime between November and January. Prior to the final warming, the SH polar vortex undergoes a downward shift in its location relative to its midwinter position (Mechoso et al., 1985). The downward shift of the polar vortex has been linked to a poleward shift of the SH eddy-driven jet in austral

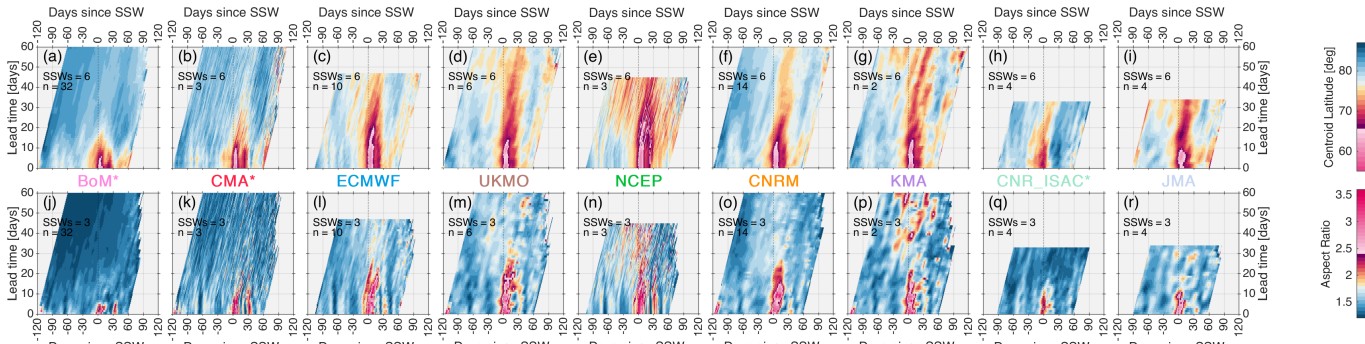

**Figure 8.** Predictions of boreal stratospheric vortex (a-i) centroid latitude (j-r) aspect ratio, computed at 10 hPa from hindcast geopotential height. Values shown are an average of all ensemble members over all (a-e, f-i) displacement and (j-n, o-r) split SSWs in the 1999-2020 record. Data are plotted against (horizontal axis) SSW-relative day number and (vertical axis) model lead time. For each panel, the number of SSWs ('SSWs') and number of ensemble members ('n') is indicated at top left. The colorbar transitions to pink occur at thresholds of 66°N for centroid latitude, and 2.4 for aspect ratio.

spring, while the timing of the polar vortex breakdown has been linked to the equatorward shift of the eddy-driven jet between November and January (Hio and Yoden, 2005; Byrne et al., 2017).

Given the above, the SH spring season can be regarded as a 'window of opportunity' for more skillful tropospheric forecasts on S2S timescales provided that stratospheric variability is accurately represented. Indeed, previous studies have shown that the SH tropospheric variability during spring, prior to the polar vortex breakdown, can be predicted from stratospheric initial

conditions in winter (Seviour et al., 2014; Lim et al., 2018; Byrne et al., 2019; Rao et al., 2020b). However, evaluations of individual seasonal prediction systems such as the ECMWF reveal unrealistic SH stratospheric variability and an inability to correctly represent stratosphere-troposphere coupling during austral spring, with likely impacts on the tropospheric mean state during that season (Polichtchouk et al., 2021).

As with the NH, we first explore biases in the wave driving of the SH stratosphere, represented by the 100 hPa eddy heat flux,

$v'T'$ (Figure 9). In the SH, negative eddy heat fluxes are poleward and represent upward propagation of wave activity into the stratosphere. In ERA-I (Fig. 9a), the poleward eddy heat fluxes are largest over the Southern Ocean with local maxima south of Australia (150E) and downstream of the Antarctic Peninsula (30W). Eddy heat fluxes climatologically peak in amplitude in austral spring and are associated primarily with stationary waves of wavenumber-1 with a secondary role from transient waves (Randel, 1988). Similar to the NH (Figure 5), the spatial patterns of eddy heat flux biases strongly differ between the high-

and low-top composites, particularly at longer lead times. At week 4, the low-top composite shows positive biases over the two local maxima in the ERA-I climatology; however, this is primarily due to the BoM system (see also Figure 9d), which has a very large positive bias in the region between Antarctica and Australia. In contrast, the pattern of biases in the high-top composite is much less coherent, except for a relatively large region of negative biases between the southern tip of South America and the Antarctic peninsula (indicating more upward wave driving). As a function of lead time (Figure 9d), the biases



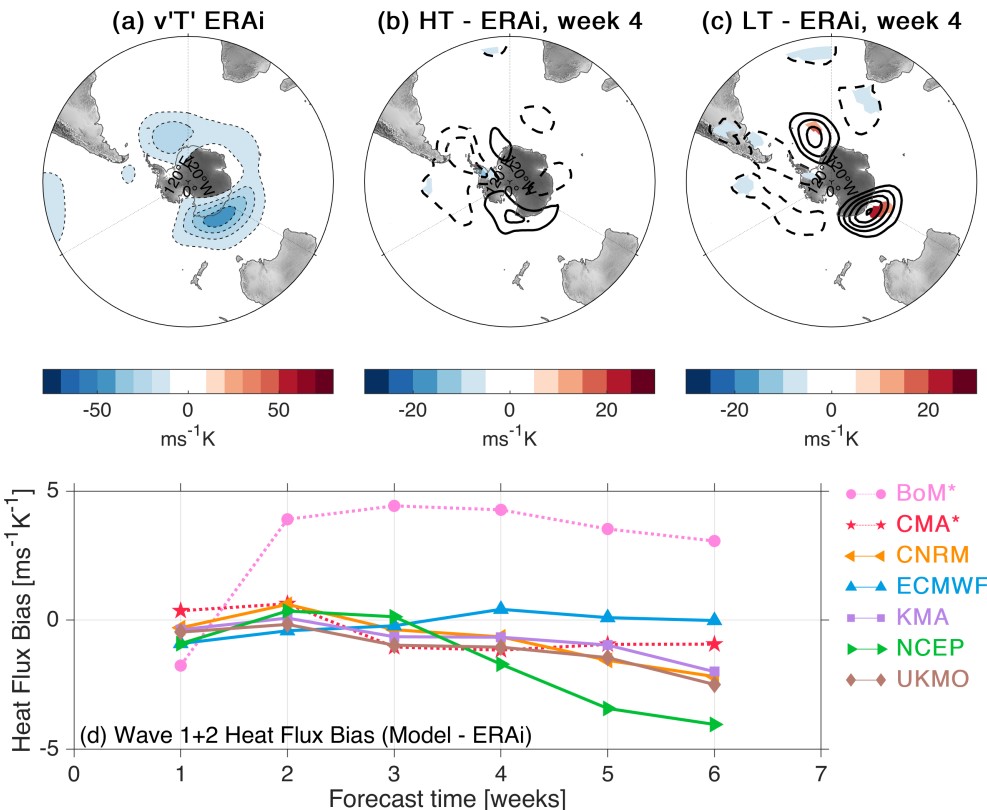

**Figure 9.** (a) The September-November climatology of ERA-I eddy heat fluxes, $v'T'$, at 100 hPa in the Southern Hemisphere. (b) The high-top composite of eddy heat flux model biases with respect to ERA-I from August-October initializations at a lead time of 4 weeks. (c) is as in (b), but composited for the low-top models. In panel a, the line and colour-filled contours match the colorbar spacing of 10 ms$^{-1}$K; in panels b and c, the line contours match the colorbar contour intervals of 5 ms$^{-1}$K, but colours are only shown where the biases are statistically significant at the 95% level from a two-tailed Student's t-test. (d) Time series of the difference between the S2S hindcasts and ERA-I for the combined wave-1 and 2 planetary wave heat flux over 45°-75°S, based on August-October initializations. Note that the low-top composite in panel (c) includes results from the CNR-ISAC system for a total of 3 low-top systems in the composite, which is not shown in panel (d) (only 2 low-top systems).

among the different systems generally become more amplified beyond 3 weeks, with most of the systems showing too negative heat fluxes (i.e., predicting enhanced poleward eddy heat flux), particularly at weeks 5 and 6. Among these, NCEP is the clear outlier with the most negative heat fluxes. The fact that the high-top systems seem to slightly over-estimate SH upward wave driving could imply that these systems also have too high springtime variability in the SH stratosphere.

Following on from the SH eddy heat flux biases, Figure 10 shows the interannual standard deviation of SH polar cap (60°S-90°S) geopotential height at 50 hPa for each of the S2S systems based on initialization dates that are closest to the first of August, September, October, and November. Here we consider the common hindcast period of 1999-2010, but exclude 2002, the year of the only major sudden stratospheric warming in the SH. In ERA-I, the observed interannual variability increases



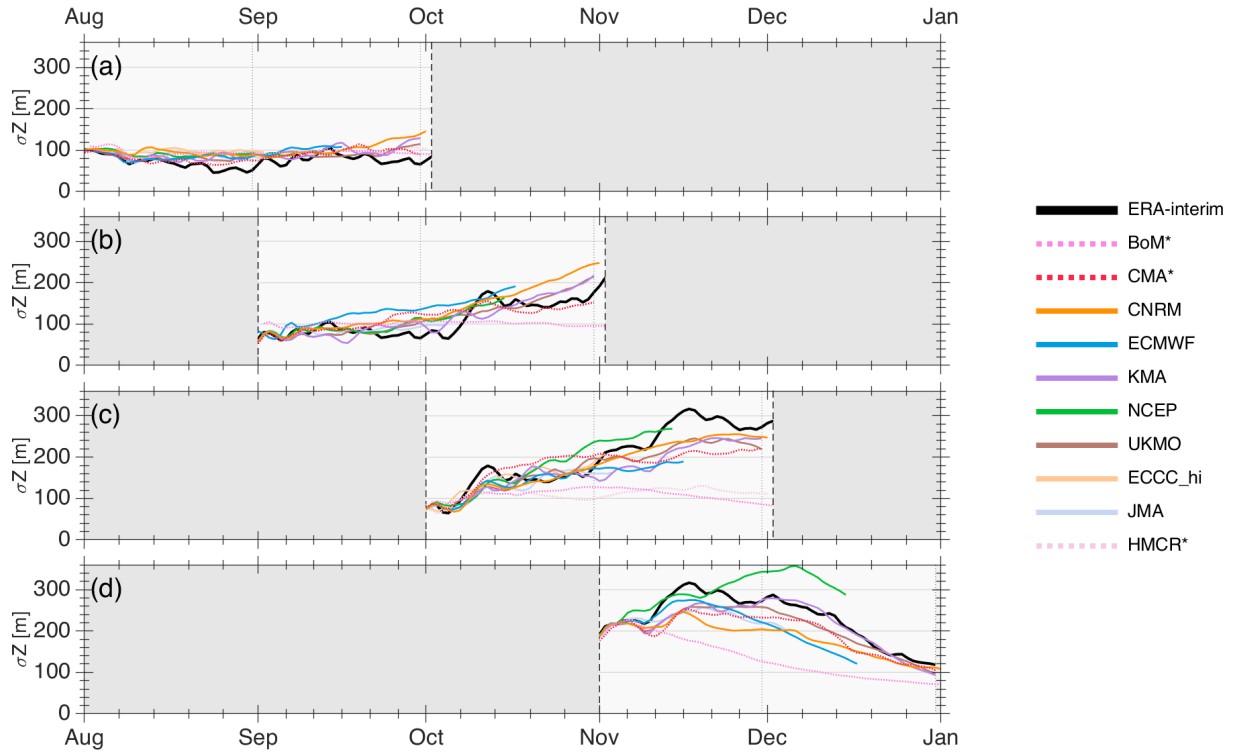

**Figure 10.** Interannual standard deviation of the southern hemisphere hindcast 50 hPa polar cap (60°-90° S) geopotential height for each model (coloured lines) in comparison to ERA-I (black lines). The hindcasts closest to the first of each month are chosen for each model. The verification time range in each row spans a maximum of 8 weeks.

considerably from the beginning of October, peaking in mid-November and decaying thereafter. The hindcasts initialized in August and September present similar levels of variability as reanalysis up to early October; for initializations in early September, several models (CNRM, ECMWF, KMA, UKMO) even show an increase in variability at longer lead times. However, the difference between the models and observations increases for initializations in early October and November, with most models underestimating the observed variability beyond 4 weeks. For early November initializations, most of the high top models (with the exception of NCEP) underestimate the variability at shorter lead times from roughly week 2 onwards. Clearly though, the low-top models show the most consistent underestimation across the different months and lead times; BoM and HMCR particularly show nearly flat variations, indicating that they do not simulate an appreciable seasonal cycle in variability. The overestimation of SH variability shown by NCEP from week 4 and beyond in October and November initializations is consistent with this system having the most negative heat flux biases in Figure 9. Overall, because of the limited number of years in the comparison, most of these differences in variability with respect to reanalysis are not significant (not shown).

There are also biases in the timing of the seasonal breakdown of the SH polar vortex. Figure 11 shows histograms of the polar vortex breakdown dates across different initialization dates between August and November. Here we simply define the

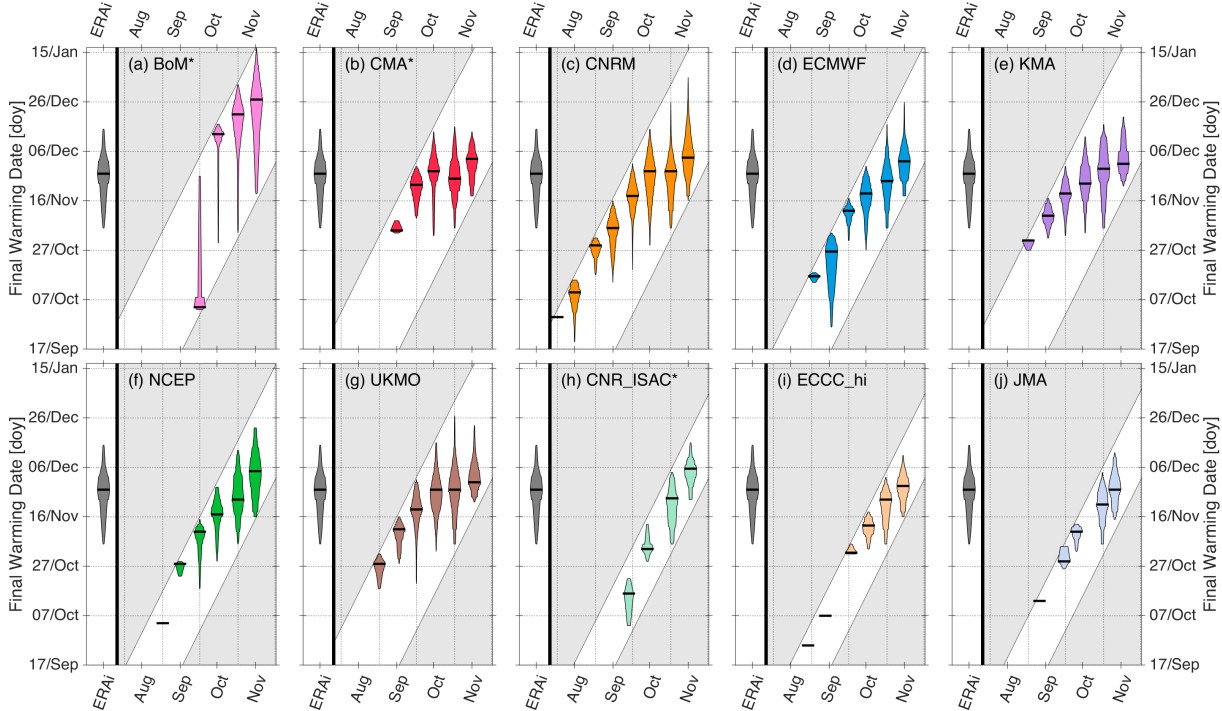

**Figure 11.** Violin plots showing the distribution of southern hemisphere final warming dates (shown as the day of year) for each S2S system. The distribution of final warmings from ERA-I is shown in the grey violin in each panel. Each distribution represents a composite of hindcasts initialized closest to the beginning or middle of each month. The diagonal white region represents the approximate maximum forecast horizon for each system.

breakdown date as the first day of easterlies without subsequent return to westerlies at 10 hPa and 60°S. For many of the models, forecasts initialized in early spring produce easterly winds towards the end of the forecast, but at times that are much too early for the breakdown. This is a somewhat surprising result given that such early polar vortex wind reversals are rare in observations and expected to be rare in model simulations (e.g., Jucker et al., 2021). It is unclear, however, whether this

behavior represents an early breakdown bias among some of the forecast systems, or whether these events represent SSWs for which the polar vortex would eventually recover if the forecasts continued in time. Regardless, these models produce early events that are generally not consistent with the observational record. Breakdown dates that fall within the ERA-I range are generally only produced for initializations on or after the first of October, or once the end of the forecasts include the second half of December. An exception is the BoM model, which produces breakdown dates that consistently fall close to the end of

its forecasts, resulting in a late breakdown bias for initializations after October. Notably, free-running coupled climate models have a bias towards too-late breakdowns of the SH polar vortex (Butchart et al., 2011; Rao and Garfinkel, 2021b); this suggests that information contained in the October initializations likely help to constrain the S2S models and improve final warming estimates. Although stratospheric ozone is prescribed to climatological values in many S2S forecast systems, the strength




of the initialized polar vortex winds in October likely contains information about relevant chemistry-climate feedbacks with
stratospheric ozone that are well-correlated to the timing of the breakdown date (Butler and Domeisen, 2021).

## 4 Discussion and Conclusions

We have performed a comprehensive intercomparison of stratospheric biases in subseasonal forecast systems, with a core focus
on systems that contribute to the S2S database (Vitart et al., 2017). Our results show:

- Forecast systems with low-top atmospheric models generally have the largest biases across the diagnostics examined for
zonal mean winds and temperatures, the QBO, meridional eddy heat fluxes, and the stratospheric polar vortices.

- Global- and annual-mean warm biases in the stratosphere tend to be most common across the different S2S forecast
systems, though this can vary for different regions of the stratosphere in some (e.g., lower versus middle stratosphere).

- Too strong/cold wintertime polar vortices, and too cold extratropical UTLS regions are common features across most of
the systems in the zonal mean temperature and zonal wind biases.

- Tropical stratospheric anomalies associated with the QBO tend to decay toward climatology with lead time. For high-top
systems, this issue appears to be worse in the lower stratosphere, and relatively worse for EQBO phases than WQBO.

- In the NH, most S2S forecast systems do not capture the seasonal cycle of extreme vortex event probabilities; for
example, the occurrence of SSWs and strong vortex events are underestimated and overestimated, respectively, for weeks
3-4 forecasts initialized in January. Similarly, the S2S systems generally underestimate the magnitude of wind changes
associated with observed SSWs and strong vortex events, even at lead times within 2 weeks.

- In the SH, most systems generally underestimate the late-spring variability of the Antarctic polar vortex, particularly for
initializations in October and November. However, many systems also simulate reversals in the 10 hPa 60°S zonal mean
zonal winds for initializations in August and September, at times of the year when SH final warmings have not occurred
in reanalysis.

These biases likely arise due to a combination of factors. While the physical processes that govern the evolution of the
stratosphere are relatively well understood, they are generally not fully resolved within atmospheric models, and are instead
dependent upon model configurations (e.g., the height of model lids and vertical resolution), and simplified/parameterized
processes (e.g., gravity wave drag, and the representation of ozone). The resulting biases can affect both the mean state and
variability of the stratosphere, and have potential consequences for subsequent coupling with the troposphere.

Several of the systems with the largest global- and annual-mean warm biases in the stratosphere are those in the "NOAA
family", including the high-top GEFSv12 and NCEP CFSv2, and the low-top GFDL-SPEAR; the others include the low-top
BoM and CESM2-CAM. The only systems with global- and annual-mean cold biases are the ECMWF and CESM2-WACCM
systems. It is unclear whether the warm-biases in the NOAA systems are related to a common cause. While the GEFSv12 and





NCEP CFSv2 models use similar physics packages, including ozone physics parameterizations and radiation packages, they
do use different dynamical cores (Guan et al., 2021; Saha et al., 2006, 2014); similarly, GEFSv12 and the GFDL-SPEAR share
the same FV3-based dynamical core, but SPEAR uses different physics packages and uses prescribed monthly ozone time
series (Zhao et al., 2018; Delworth et al., 2020). The fact that the global and annual-mean stratosphere should be in radiative
equilibrium poses a strong constraint that biases are likely to be radiative in nature, but model dynamics related to horizontal
and vertical resolution can also play a role, particularly at high resolutions (see, e.g., Polichtchouk et al., 2019). We note that
the global mean cold biases in the ECMWF system are likely dependent on the specific model cycles, and that more recent
versions likely have reduced cold biases following implementation of quintic vertical interpolation in the ECMWF model's
semi-Lagrangian numerics (Polichtchouk et al., 2019).

The cold biases in wintertime polar cap temperatures (corresponding to stronger polar stratospheric winds), and cold biases
in the extratropical upper troposphere/lower stratosphere are longstanding biases that are similar to what has been documented
in other weather and climate models (Charlton-Perez et al., 2013; Bland et al., 2021). The stratospheric polar cap temperature
biases generally point to a dynamical cause related to parameterized gravity wave drag, given that wave-mean flow interactions
and the ensuing residual circulations are responsible for driving local zonal mean temperatures away from radiative equilibrium.
The cold extratropical UTLS biases, on the other hand, are likely to be radiatively driven, related to excessive leakage of water
vapor into the lower stratosphere (e.g., Bland et al., 2021). Both of these issues are dependent upon vertical resolution, which
likely explains why the biases in the low-top systems (which have fewer levels in the stratosphere and coarser resolution in
the UTLS) are generally more severe than those in the high-top systems. Correcting such biases is likely to have some impact
on forecast skill in both the stratosphere and troposphere, since the latitudinal dependence of the temperature biases affect
the distribution of winds (i.e., the tropospheric jets and stratospheric polar vortices). For instance, artificially bias-correcting
the extratropical UTLS humidity biases in the ECMWF system was shown to remove the UTLS cold biases, and moderately
improve the skill of forecasts over Europe (Hogan et al., 2017).

The gradual decay of QBO anomalies with lead time towards each forecast system's own climatology is consistent with
possible issues related to parameterized gravity wave drag. In models, the QBO has been shown to be most sensitive to pa-
rameterized non-orographic gravity wave drag (NOGWD) (e.g. Bushell et al., 2020), but other factors such as model vertical
diffusion and resolution can also play a role in properly representing and maintaining the QBO, especially in the lower strato-
sphere (Garfinkel et al., 2021; Polichtchouk et al., 2021). However, documenting the specific model configurations and gravity
wave parameterizations among the different forecast systems examined herein is beyond the scope of this study. Our results
also showed that many of the S2S systems show a "Holton-Tan" response to the QBO wind phase in polar vortex winds and
polar cap geopotential heights consistent with observations, but only for the first 2-4 weeks of the forecasts, after which the
polar vortex anomalies decay.

To better understand the origin of stratospheric polar vortex biases, we examined the distribution and time evolution of
NH lower stratospheric meridional eddy heat fluxes, which represent upward wave driving. These heat fluxes were generally
more realistic in the high-top systems; the low-top systems showed considerable negative biases in heat fluxes at week 4
over the Northern Pacific and Scandinavia/Siberia. This suggests that, in combination with their limited representations of the




stratosphere, these low-top systems have difficulties simulating realistic Rossby wave activity in the troposphere and/or their
propagation and interaction with the mean flow (see, e.g., Schwartz et al., 2021). Thus, other biases documented for the low-top
systems (such as those related to polar vortex winds and variability) are likely tied, to some extent, to these deficiencies. We
documented similar behavior in the low-top composite of SH eddy heat fluxes, but the biases for the low-top systems were less
robust, and were particularly affected by the BoM system having large positive heat flux biases (indicating less upward wave
driving in the SH).

Raw forecasts from the S2S systems do not accurately capture the seasonal cycle of boreal SSWs or strong vortex events. Re-
moving the drift in the stratospheric zonal winds through simple bias-correction does improve the probabilities for these events,
but primarily only for midwinter occurrences. While perfect prediction of such extreme polar vortex events is not feasible, ide-
ally the statistics for their monthly occurrences derived from the hindcasts would more closely match those from reanalysis,
especially at longer lead times and assuming similar levels of tropospheric "noise". Furthermore, the underestimation of the
magnitude of wind changes that occur surrounding extreme polar vortex events (even within 1-2 weeks of lead time) suggest
that the S2S systems would likely have issues simulating downward coupling associated with such events. The persistence and
magnitude of SSWs and strong vortex events is thought to be an important aspect that helps determine whether they lead to
coupling with the troposphere (Maycock and Hitchcock, 2015; Karpechko et al., 2017; Charlton-Perez et al., 2018; Domeisen,
2019; White et al., 2020).

In the SH, a large fraction of the S2S systems seem to simulate early breakdowns of the SH polar vortex at 10 hPa, even for
forecasts initialized in August and September. Because the forecasts are truncated in time, we cannot say whether these would
be considered SSWs or final warmings, but nonetheless our results reveal that false-positive easterly wind events are relatively
common in the S2S hindcasts, which is not a phenomena seen in observations.

Our results show that many of the mean state biases become more sizable with increasing lead time (as expected), especially
around weeks 3 to 4. Predictions that in some way rely on stratosphere-troposphere coupling processes are unlikely to be
meaningfully affected by the small biases present at shorter lead times of 1-2 weeks. For example, the failure of a model to
predict a SSW beyond 2 weeks is unlikely to be related to the model's tendency for its initial EQBO anomaly to decay by 2-3
m/s in the first few weeks (Figure 3). However, many of the S2S systems considered herein make forecasts well beyond 4 weeks;
on these extended timescales, such biases are likely to be more important. Furthermore, while a fully unbiased model will not
be possible to achieve, it is still desirable for models to (1) minimize mean state stratospheric biases so that the stratosphere
represents a more accurate upper boundary condition for the troposphere, and (2) have similar variability (in a statistical sense)
to observations in the stratosphere so that ensemble spread properly accounts for potential outcomes. For example, forecasts
from a model with a strong NH polar vortex bias could simply be post-processed with bias correction to improve the prediction
for a SSW (Figure 6); however, if the polar stratospheric winds stay westerly in the actual model simulation, then that would
represent a different dynamical regime for stratosphere-troposphere coupling compared to the model actually simulating a
transition to stratospheric easterlies (since easterly winds effectively shut off vertical propagation of Rossby waves).

This study primarily focuses on biases among S2S models within the stratosphere. We have shown that large biases are
present throughout the stratosphere, linked to a range of stratospheric phenomena in the tropics and the extratropics of both



hemispheres. To our knowledge, this is the first systematic assessment of such biases in the stratosphere and of processes
affecting the stratosphere in a multi-model study of sub-seasonal to seasonal prediction systems. In a follow-up companion
study as part of the same SNAP effort, we more closely examine how biases in the stratosphere, such as those identified herein,
are linked to coupling with the troposphere and its predictability.

*Data availability.* The hindcasts from the S2S database used here are available from https://apps.ecmwf.int/datasets/data/s2s/ under the
"Reforecasts" S2S set. The NOAA GEFSv12 hindcasts can be obtained from https://registry.opendata.aws/noaa-gefs-reforecast/. Hindcasts
for CESM2-CAM are available at https://www.earthsystemgrid.org/dataset/ucar.cgd.cesm2.s2s_hindcasts.html, while those for CESM2-
WACCM are from https://www.earthsystemgrid.org/dataset/ucar.cgd.cesm2-waccm.s2s_hindcasts.html. Data for the GFDL-SPEAR can be
made available upon request.

*Author contributions.* ZDL organized and led the SNAP effort leading to this paper, with help from CIG and AHB. ZDL, AHB, CIG, and
DIVD drafted the manuscript. CJW compiled all the figures in the paper and supplement with data provided from the following collabora-
tors/analyses: HK, and S-WS performed the analyses for Figures 1 and 2. CIG and CS performed the analyses for Figures 3 and 4. CIG,
CS, FMP, JG-S, AdlC, and NC performed the analyses for Figures 5 and 9. SHL performed the analysis for Figure 6. RW-YW and DIVD
performed the analysis for Figure 7, including with data for the GEFS and CESM2 systems provided by ZDL. NPH and CJW performed
the analysis for Figure 8 with code provided by ZDL. MO, MJ, and IP performed the analyses for Figures 10 and 11. These analyses were
all done for the systems provided in the S2S database; ZDL performed the same analyses for Figures 1, 2, 3, and 6 for the GEFSv12 and
CESM2 systems. PL and LJ performed the same analyses for Figures 1 and 2 for the GFDL-SPEAR system. All the listed coauthors were
active participants in this SNAP community effort and provided comments on the draft manuscript.

*Competing interests.* Daniela Domeisen is a member of the editorial board of Weather and Climate Dynamics.

*Acknowledgements.* This work uses S2S Project data. S2S is a joint initiative of the World Weather Research Programme (WWRP) and the
World Climate Research Programme (WCRP). This work was initiated by the Stratospheric Network for the Assessment of Predictability
(SNAP), a joint activity of SPARC (WCRP) and the S2S Project (WWRP/WCRP).

The work of R.W. is funded through ETH grant ETH-05 19-1. Support from the Swiss National Science Foundation through projects
PP00P2_170523 and PP00P2_198896 to D.D. is gratefully acknowledged. C.I.G. and C.S. are supported by the ISF-NSFC joint research
program (grant No.3259/19). The work of M.O. was supported by UBACyT20020170100428BA and PICT-2018-03046 projects. The work
of A.dlC. is funded by the Spanish Ministry of Science through project PID2019-109107GB-I00. B.A and N.C. acknowledge the support
of the Spanish Ministry of Science and Innovation through the JeDiS (RTI2018-096402-B-I00) project. FMP and JG-S have been partially
supported by the Spanish ATLANTE project (PID2019-110234RB-C21) and "Ramón y Cajal" programme (RYC-2016-21181), respectively.
N.P.H. and C.J.W. are supported by UK Natural Environment Research Council (NERC) grant number NE/S00985X/1. C.J.W. is also





supported by a Royal Society University Research Fellowship UF160545. S.-W.S. and H.K. are supported by the Basic Science Research Program through the National Research Foundation of Korea (2017R1E1A1A01074889). This material is based upon work supported by the

U.S. Department of Energy, Office of Science, Office of Biological & Environmental Research (BER), Regional and Global Model Analysis (RGMA) component of the Earth and Environmental System Modeling Program under Award Number DE-SC0022070 and National Science Foundation (NSF) IA 1947282. This work was also supported by the National Center for Atmospheric Research (NCAR), which is a major facility sponsored by the NSF under Cooperative Agreement No. 1852977. P.L. is supported by award NA18OAR4320123 from the National Oceanic and Atmospheric Administration (NOAA), U.S. Department of Commerce. ZDL was partially supported under NOAA Award

NA20NWS4680051; ZDL and JP also acknowledge support from US Federally Appropriated Funds. The statements, findings, conclusions, and recommendations are those of the author(s) and do not necessarily reflect the views of NOAA, or the U.S. Department of Commerce.



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
