# Peer review of "Quantifying stratospheric biases and identifying their potential sources in subseasonal forecast systems"

_Weather and Climate Dynamics, 2022_

## Author Response (AR1)

**Reviewer #1**

A very thorough and useful study into the stratospheric biases present in the current seasonal forecast systems. I note that a further study is planned to detail how these biases might impact forecast skill, and I think this paper will nicely underpin future research in that area. A couple of minor points:

We thank the reviewer for their kind comments and their careful consideration of our manuscript. We have made changes to text and figures following the reviewer's suggestions, which we think has improved the manuscript. Below we address each individual comment in blue font. Any reference to line numbers refers to the clean manuscript without tracked changes.

1) Line 153 states "8 high-top and 6 low-top models". Yet there are 9 high-top and 6 low-top models in Table 1, and 9 high-top and 5 low-top models in Figure 1 (ECCC-low is in Table 1 but not Figure 1). I suspect the text should read "9 high-top and 6 low-top models", and then give a reason for ECCC-low not being included in Figure 1.

Thank you for catching this typo. We have corrected the text. We do not give a specific reason why ECCC-low is not included in Figure 1, since we note that the precise number of models included in a given analysis varies. However, we did notice that we do not actually mix ECCC-low and ECCC-high in any of our figures, so we have also corrected the text saying as such (please see L133-135): "In contrast, ECCC data is considered to be optional, ***so some analyses use either the high- or low-top versions***; in such cases, we explicitly describe these as ``ECCC-hi'' or ``ECCC-lo'', respectively."

2) Line 197 notes "apparent differences" between high and low top models. Please state whether these differences are statistically significant. Actually this point is also relevant to Figure 2 -- please add stippling to Figure 2 to show the regions where the high-top and low-top model biases are significantly different from each other.

In Figure 1 we have added 2 columns on the end of the figure showing the high and low top composite biases for each pressure level, with bold text denoting statistically significant differences (italicized not significant). We have added a sentence to the text discussing these results; essentially the systems are only significantly different at 50 hPa, which we think is likely due to the caveats we note in the text (specifically, the fact that some of the high top systems have relatively large biases on par with the low top systems). Please see L204-206:
***"Annual-mean global-mean temperature differences between the high- and low-top composites are thus only significant at 50 hPa (numbers in bold), where the low-top systems have only positive biases, and the high-top systems have mostly slight negative biases"***

[Figure]

**Figure R1:** Updated version of Figure 1 in manuscript, including the HT and LT composites and assessment of significant differences in the last two columns.

For Figure 2, we have added line contours showing the mean absolute errors (MAE) in response to the other reviewer's comments. We tried adding both the stippling and MAE contours, but thought it made the figure a little too busy. Thus we added a figure to the supplement showing the high- minus low-top composite differences, with stippling overlaid (see Figure S15). The stippling shows some of the patterns we discuss – significant differences in the global mean temperatures at 50 hPa (consistent with Figure 1), in the NH DJF polar vortex winds and temperatures at 10 hPa (somewhat difficult to see because of the figure being limited to 10 hPa), and in the UTLS temperature biases at 200 hPa for the SH DJF. We have modified the text discussing Figure 2 to include some additional sentences/discussion regarding both the MAE and the significant differences.

3) Line 217: You note that KMA is not used in the high-top composite as it is very similar to UKMO. But then you are not using all the information that you have. I would suggest that, if these models really are very similar, you treat them as a single ensemble -- i.e. combine the ensemble members from UKMO and KMA, and then form an ensemble mean from those. That way you use more information (we know increased ensemble size is good) and still avoid biasing your composite.

Thanks for this suggestion – we actually agree that this would be a great idea for some kinds of analyses (particularly those considering predictability). However, with our focus on biases, we prefer to show the different systems explicitly, since we think it is useful for readers to see how similar the UKMO and KMA systems are across the different analyses we consider.

In this specific case, the zonal mean biases for individual systems are shown in the supplement; if we combined UKMO and KMA, the zonal mean bias of the combined ensemble would be very similar, and thus the high top composite would not be affected much.

4) Line 380: I assume the magnitude of SSWs is computed for each ensemble member, and then averaged? If you are looking at zonal wind changes in the ensemble mean then the changes may be too small simply due to different central dates in different ensemble members.

Yes, this is correct – the zonal wind changes are computed for individual ensemble members and then averaged. We have added a sentence in the manuscript that clarifies this (please see L394-397): "The deceleration or acceleration associated with these events is measured by computing the change in the hindcast/reanalysis zonal-mean zonal wind at 10 hPa and 60N, at ±5 days around the ERA-I event onset dates; *for the hindcasts, these are first computed individually for each ensemble member before being composited.*"

Reviewer #2

The study by Lawrence et al provides a comprehensive evaluation of the models contributing to the subseasonal to seasonal (S2S) hindcast database with respect to their representation of the stratosphere. This evaluation exercise is relevant because there is increasing focus on analyzing the impact of the stratosphere on (sub-)seasonal predictability; the evaluation of the stratosphere-troposphere coupling, which is in this context even more relevant, is planned for a follow-up paper by the group of authors. A number of common biases are identified here, often consistent with long-standing biases known from climate model evaluations. As such, the study provides a useful compendium of stratospheric diagnostics that both users of the data as well as modeling centers can refer to in future applications and model development. The study mostly falls short to reveal the (possible) reasons for the biases, but this is a task that cannot be expected from such an evaluation effort. Overall, the study is well written and I can recommend publication after some (mostly minor) issues are addressed, as detailed in the specific comments. In particular I have concerns with some of the diagnostics and/or their interpretation, that should be addressed by the authors.

We thank the reviewer for their comments and for pointing out portions of our manuscript that required more careful attention. We have made changes to the paper and figures that we think have helped to improve the manuscript's overall clarity. Below we address each individual comment in blue font. Any reference to line numbers refers to the clean manuscript without tracked changes.

Specific comments:

- page 5, line 128: The comment on excluding the newer cycles of the ECMWF model is confusing - as such the modification of the model would rather be a motivation to include the data in order to assess its impact on the stratospheric representation. I guess this has to do with avoiding to mix different model cycles, and not having the data available for the relevant periods, but this could be explained better in the text.

Thanks for pointing this out. As you note, we wanted to cover the 1999-2010 period while not mixing cycles with large differences in the prediction systems. We have clarified this text to read (see L127-130): *"In order to cover the 1999-2010 hindcast period, for ECMWF we consider only CY43R1, CY43R3, and CY45R1 to prevent mixing cycles with large changes to the prediction systems; cycles beyond CY46R1 were excluded because these hindcasts were initialized with ERA5 reanalysis, include updates that explicitly affect mean stratospheric biases (citation), and do not fully cover the 1999-2010 period."*

- page 7, line 175: Why "approximately" the 80th percentile, but still this is a very specific value (41.2 m/s) ? I would expect to either use a rounded value ( 40 m/s), which is "approximately" the 80th percentile, or this value is indeed very close to the 80th percentile, in which case you can drop the "approximately".

Good point – we have removed the word "approximately" and maintained the use of the 41.2 m/s threshold to remain consistent with the other studies cited.

- page 9 / Figure 2: I wonder whether the averaging over the groups of high- versus low-top models makes much sense - E.g. in the group of the high-top models, there are ~4 models with global mean warm biases (dominated by GEFSV12 and NCEP), and ~4 models with (less strong) cold biases at 10 hPa, so in the average there is some compensation going on, and the total bias is likely dominated by the two models with strongest biases (GEFSV12 and NCEP). Also in the group of low-top models, there is some compensation between models with strong warm and cold biases. So excluding a specific model with a strong bias from the composite of models would likely strongly change the picture, thus making the analysis somewhat meaningless. If the aim is to show that the group of high-top models generally has smaller biases (plus reveal the regions of biases), it could be better to calculate a metric like mean of the absolute error, or a root-mean square error across the groups of models.

We understand your concern here. However, compositing high- and low-top models is a relatively standard practice of prior studies in a similar vein. Additionally, the sign of the errors (the biases) are important for the interpretation of the results. For instance, we would interpret a warm/weak vortex bias differently than a cold/strong vortex bias, as these would likely be related to too much versus too little wave driving, respectively. This kind of information would be lost by using a strictly positive metric such as mean absolute error (MAE) or root-mean square error.

While there is likely to be some sensitivity of the results to excluding some systems in the different composites, we did not make any attempt to hide such a fact since we explicitly show the zonal mean biases for individual systems in the supplement. Nonetheless, we thought it was a great idea to augment these results by adding the MAE to Figure 2 (included below this response) and the supplemental figures. Now, all these figures show the MAE in the grey line contours. As you note, these results explicitly show the high top models generally have smaller errors in both the zonal mean temperatures and winds virtually everywhere.

[Figure]

**Figure R2:** Updated version of Figure 2 in manuscript, including line contours of MAE.

- page 11, line 265: I expect the GW parameterization will be relevant here, too.

We have added gravity wave parameterization here as suggested.

- Fig. 3: Could it make sense to show the difference of a model's drift (panels a,c,e,g,i,k) minus the respective subsampled ERAI value (panels b,d,f,h,j,l). As is, the Figure requires quite a bit of comparison in the head between the panels to interpret the model differences (e.g. BoM seems to be a clear outlier, but at in some cases (e.g. U50 QBO) this seems to be at least partly due to the sampling, as also to ERAI sampled at BoM is offset to the other models).

We have modified Figures 3 and 4 as suggested so that they show the anomaly difference from reanalysis (i.e., "hindcast anomaly minus ERA-I anomaly"). These are shown below this response. We also changed these so that the anomalies are computed equivalently as "hindcast/reanalysis minus climatology" (in the submitted version of the manuscript, we instead did the WQBO composites as "climatology minus WQBO hindcast/reanalysis" so that the data in both composites could be scaled on the same y-axes and better highlight the linearity of the signals).

We have also modified relevant text in the paper to reflect the change in these figures.

[Figure]

**Figure R3:** Updated version of Figure 3 in manuscript showing the hindcast anomalies minus the reanalysis anomalies for different QBO metrics/phases.

[Figure]

**Figure R4:** Updated version of Figure 4 in manuscript showing the hindcast anomalies minus the reanalysis anomalies for the polar stratospheric quantities in different QBO phases.

- Fig. 3 /4: in both instances, the ERAI subsampled data to the BoM model appears to an outlier. I first suspected that this has to do with limited sample size, but according to the numbers in the Figures and from checking the table 1, BoM is actually the system with most members. Can you comment on the reason for the difference, and possibly add a sentence to the manuscript? Does this mean the other models are sub-sampled (but why do they tend to agree, then?)

While we changed the appearance of Figures 3 and 4 as suggested above, we did look a bit deeper at this. There are a couple of factors – one of them is that these QBO figures are based on the full hindcast periods available for each model (to maximize the number of QBO cycles, as noted in the text). For BoM, this means data going back to 1981 is included in the ERAI composites. Another big factor is related to how initializations are chosen for the composites. We subset initializations into E/WQBO based on the wind anomaly in the first 3 days of the hindcasts. In BoM, it turns out that, particularly for WQBO, the initial QBO anomalies degrade so quickly that only the strongest WQBO inits make it into the BoM composite (which impacts the ERAI T100 signal).

- page 12, line 286: I expect you mean to say that you need to consider initialization dates before the Holton-Tan effect is established, so it is not prescribed in the initial conditions, but can freely evolve in the models? As is, the sentence sounds a little confusing.

The Holton-Tan effect is still apparent in early winter, just weaker – but we see your point about how this sounds confusing. We have modified this to read (see L301-303): ***Since the Holton-Tan effect develops in early winter and is most pronounced in mid-winter, Figure 4 is limited to forecast initializations within November and December, before the effect is strongly embedded in initial conditions."***

- page 13, line 295: "generally fail": consider changing to "mostly fail" or such, since some systems (ECMWF, NCEP) do show some signature of the effect.

Changed as suggested.

- page 13, line 300: I would change the term "upward wave driving" to either just "wave driving" or "upward wave fluxes that disturb the polar vortex" or "upward wave fluxes that lead to wave driving of the polar vortex". The wave fluxes can be described as (propagating) upward, but the wave driving is not "upward". Likewise, in like 301, the meridional eddy heat flux is not a metric of the wave driving, but of the upward wave flux (being the main component of the upward Eliassen-Palm flux).

Yes, this is a good point – we apologize for the imprecise language. We have changed the wording as suggested to say "upward wave fluxes" in both cases.

- page 13, line 303: consider re-wording "such wave-driving should be resolved" to be more precise: the planetary waves, which we know are the major contributor to stratospheric wave driving, are definitely resolved in the models, also synoptic wave activity, which plays a role around e.g. the subtropical jet is also resolved. Maybe you rather mean "well represented" rather than "resolved"? Could change to e.g. "the wave driving by planetary and synoptic waves is resolved, and its representation is dependent upon... "

Similar to above, we have changed the wording as suggested to say "well represented", which is what we originally intended.

- page 14, line 306: what does "long-term coupling" refer to here? I suspect coupling on the sub-seasonal time-scales?

We have clarified this sentence to say (see L321-323): ***"Importantly, the occurrence of such polar vortex events can lead to coupling with the troposphere that lasts for weeks to months …"***

- page 14, line 309 ff / Fig. 5: The analysis of heat fluxes as a proxy of wave activity in the stratosphere is a good metric and valuable addition to simply looking at the polar vortex in order to get at the process representation. However, I have some concerns with how the diagnostics are used / discussed here. In particular, the "map" of heat fluxes reveals "two centers of action" (line 313). This pattern simply emerges because (as is well known) the wave activity is dominated by planetary waves, in particular it is wavenumber 2 that is showing up here. Therefore, I would argue that the representation of the fluxes as a map, and even more so the averaging over only one segment of the map is rather meaningless. Indeed, heat or momentum

fluxes are only meaningful when averaged over the scale of the dynamical feature (the wave) they are connected with (here the planetary wave 1 and 2), i.e. the zonal average should be considered. If the geographic information (latitude of maximum amplitude, or phase of the wave) of the wave is of interest, a better quantify would be geopotential height anomalies. Therefore, I'd strongly recommend to drop the averages over the segments, and focus on the wavenumber 1 and 2 zonal averages, as it is done in the lower panel. In this light the statement in line 327 ("have small biases near 0, indicating that on a global scale, the regional biases tend to cancel") is misleading: yes, the values over the zonal mean are smaller compared to the regional average, but this is because you average over the phases of the wave (as it should be done). Whether the values are "small" is not a question of comparing them to the values from the segments above, but how they compare to the mean value from ERAI.

We agree with your concerns here – the regional heat fluxes do not necessarily have meaning on their own, but with how they project onto the planetary-scale wave patterns. Unfortunately our original submission was not careful enough to discuss them in such a way. We have gone with your suggestion to remove the time series of the regional heat fluxes and have instead opted to show the time series for the wave-1 and wave-2 heat fluxes separately. We have, however, left the maps in the figure, but we have modified the text to more carefully discuss how the biases over specific regions are consistent with the wave-1 and -2 time series. We have made a similar change to the SH heat flux figure and discussion. The updated Figure 5 is included below.

[Figure]

**Figure R5:** Updated version of Figure 5 in the manuscript. The regional time series were replaced with wave-1 and wave-2, and the regional boxes were removed from the maps.

- page 16, line 351: As stated, in particular for the strong vortex events, much of the deviations might arise from the fixed threshold. I wonder whether this metric then rather measures the mean vortex biases then the ability of a model to produce extremes, in particular for the strong

vortex events (for SSW it is different, because a "0 m/s" threshold is a dynamically meaningfull value - it inhibits planetary wave propagation. For a strong vortex, no such fixed dynamical threshold exists). Have you considered using a model-dependent threshold that might be based on a specific percentile?

We agree that the threshold for strong vortex events is less physically meaningful than that for SSWs. However, the figure also shows the probability of these events after mean-state bias correction (the black lines on each bar). Your comment that "this metric then rather measures the mean vortex biases" is only true for the non-bias corrected winds (and would also apply to the SSW analysis, regardless of the additional dynamics at play). We considered the approach of showing the actual wind speeds equivalent to the 80th percentile in the models and reanalysis for different lead times/initializations, but this does not capture the same variability as the 41.2 m/s threshold (defined as the 80th percentile across NDJFM), and is thus not directly comparable with prior studies that have assessed such strong vortex events (particularly those of Domeisen et al. 2020). Assessing the probabilities with fixed thresholds allows us to test for any skew in the distribution which is not removed by a mean-state correction – and therefore the ability of the model to reach extremes equivalent to extremes in the reanalysis. In this case, we have decided to leave the analysis+figure as we originally presented it.

Note also that in the case of the bias-corrected results, the 0 m/s threshold may also not be physically meaningful, since the model dynamics evolve according to the real winds, and thus the bias-corrected winds could be below 0 m/s while the model winds are still above 0 (as we discuss in the text).

- page 18, line 383: I think you have to be careful here with the formulation on statements on prediction of vortex events: Yes, "the prediction systems are generally not forecasting extreme vortex events..." at lead times of 3-4 weeks - and they shouldn't, because we know that the predictability horizon of such events is around 2 weeks (or less) due to the chaotic nature of the atmosphere (and in particular the high non-linearity around vortex events), as you state in one of the next sentences. Consider rewording to make it clear from the beginning that this is not a shortcoming of the models, but an expected result given the nature of the system.

This is a good point – thank you for this suggestion. We have modified this text as follows (see L399-404): *"At weeks 3 and 4 lead times, the predicted wind change distributions among the systems are generally close to zero and exhibit small spread. This indicates that these models predict climatological zonal-mean zonal wind values or only weak wind tendencies of the same sign as the events. This is not a shortcoming of the prediction systems, but rather to be expected given that the typical predictability limit for these vortex events is about two weeks. Even within a lead time of two weeks, some systems still underestimate the magnitude and spread of the observed wind changes …"*

- page 18, line 395 ff.: I wonder whether the analysis of the vortex geometry adds much value to the already comprehensive evaluation. The paper is already very long, and this parts appears to add little to the understanding / quantification of relevant biases, and as stated in the last sentence, the sample size is maybe too small to make any robust statements. I also do not

understand the reasoning given in line 396, in that the "shape or location would affect vertical wave propagation" - isn't it the other way round, i.e. the waves themselves lead to distortions of the shape and location? (In the absence of wave activity, wouldn't the polar vortex be a circular feature, with its location simply given by the radiative constrains on the maximum temperature gradient?). Therefore, my suggestion would be to consider to move this part to the appendix.

After consideration, we have decided we would like to keep this figure. We think that changing the NH heat flux figure to show the wave-1 and wave-2 heat fluxes separately (see response above) provides better motivation for this figure. For example, the negative biases in the BoM and CMA heat fluxes are consistent with the poor vortex geometry predictions by these systems; similarly, the positive wave-1 heat flux bias in the NCEP system is consistent with it overpredicting lower centroid latitude values at longer leads (indicative of vortex displacements).

We have revised the confusing statement about the vortex geometry and waves to read (see L412-413): *"The shape and location of the polar vortex are ultimately affected by the vertical wave activity that influences the occurrence and/or magnitude of SSWs."*

- page 20, line 434 ff, Fig. 9: does the composition of low- and high-top models make sense here, given the limited number of low-top models (two), which is completely dominated by the BoM model (as stated in line 441)?

Because we had different coauthors handling different portions of analyses, this figure was slightly inconsistent in our first submission. The map for the low-top composite actually included 3 models (including CNR-ISAC) that were not reflected in the time series, which only showed 2 low-top systems. We have since made the figure fully consistent so that the time series for CNR-ISAC are also displayed. We also made this figure fully consistent with its NH counterpart in Figure 5 (see below) and included the wave-1 and wave-2 heat flux bias time series.

While the composites are imbalanced (3 low versus 5 high), the time series of biases for CNR-ISAC show that it does have large biases similar to BoM, particularly for wave-1. The updated Figure 9 is included below this response.

[Figure]

**Figure R6:** Updated version of Figure 9 in the manuscript. The figure has now been made to look like its NH counterpart (shown above), and includes CNR-ISAC in (d) and (e), which was already included in the composite for (c).

- Fig. 11: The analysis of the final warming date in the SH is certainly an important quantity to consider, but I have to admit I don't fully understand the analysis presented in Fig. 11. In particular I guess it comes down to the question whether the "violins" are weighted in some way by the fraction of members that do predict a final warming in the given time frame. E.g. as is written, some of the early initialization dates do already predict final warmings, but as far as I can see it is impossible to tell from the Figure how many those would be. Likewise, I suspect that for the ini. dates in mid-November, for a number of instances the final warming has already happened. So overall it is, if I'm not mistaken, not possible to infer from the Figure at which times the model most likely predicts a final warming. Maybe some scaling of the "violins" by the relative number of members that do predict a final warming at this ini. date could solve this. Further, in the caption you could expand the sentence "... initialized closest to the beginning or middle of each month" with ", given on the x-axis" or so, otherwise this information has to be guessed by the reader.

We apologize for this analysis/figure being unclear. We have moved away from displaying the data as violin plots and have instead opted to show these as "strip" plots (see the updated figure 11 attached below). These should explicitly show the discrete ensemble member predictions of the final warming dates, and we have explicitly listed the number of reversals predicted for each initialization. We have also modified the figure caption to appropriately describe the new figure and have added a statement about the timing of the initializations as suggested, which reads

 ***"The systems are composited based on initializations that fall closest to the beginning or middle of each month as specified on the x-axis."***

[Figure]

**Figure R7:** Updated version of Figure 11 in the manuscript. The figure has been changed from showing violins to showing "strips" of each individual prediction of a reversal.

- Page 28, line 520ff: I was a bit surprised by the statement that the polar cap temperature bias is specifically assigned to param. gravity waves (only). What leads you to this statement, why not planetary waves?

Yes, this was an oversight, especially since we showed some systems have notable biases in the eddy heat fluxes. This sentence now reads (see L542-545): *"The stratospheric polar cap temperature biases generally point to dynamical influences related to planetary wave drag and parameterized gravity wave drag, since wave-mean flow interactions and the ensuing residual circulations are responsible for driving local zonal mean temperatures away from radiative equilibrium."*

- page 26, line 560ff: As stated above, the number/percentage of those early final warmings from the present analysis is not clear to me (indeed the sample size is, I think, not mentioned in this specific analysis?) - possibly it is not in contradiction with observations, given that the sample size of modern satellite observations is only ~ 40 years (with essentially 2 "early final warming", counting 2002 and 2019).

Our earlier comment on this Figure addresses the changes we made to it, which we hope makes things more explicit. We think our statements remain fair, given that some of the systems

predict 10 or more reversals occurring in September/October. For instance, if we use the rough approximation of 1 event in 20 years, and treat the individual ensemble members of CNRM independently for each initialization in August, we would have 165 realizations (15 members * 11 years), for which we would expect about 8 events. Compare this to the 38 or 29 indicated for August hindcast inits.

- page 26, line 566: As above, I'd caution the authors on the formulation here - "the failure of the model to predict SSW beyond 2 weeks" implies that the model should be able to predict this, which is (to the best of our knowledge) not true.

After some discussion among the collaborators, we decided to remove the sentence including this statement since the example we gave was somewhat speculative.

---

## Author Response (AR2)

**Note on Correction to wcd-2022-12**
Zachary D. Lawrence (ZDL)

To whom it may concern,

I would like to make readers aware that there was a minor error in the submitted versions of the manuscript involving the GEFSv12 forecast system. In the original submission and revised submission, the GEFSv12 was incorrectly listed and used as a "high-top" system with a model top at 0.1 hPa. In fact, GEFSv12 has a model top at 0.2 hPa, and therefore based on the classification used in the manuscript, it should have been displayed and used as a low-top system. The origin of this error came from my (ZDL's) misinterpretation of Figure 4 in Hamill et al., 2022.

We have fixed this mistake before the final publication of the manuscript so that GEFSv12 is correctly used in low-top composites and displayed as a low-top model with dashed-lines and/or asterisks next to its name. This has resulted in only very minor changes to the manuscript text and figures, with no impact to conclusions drawn. These are briefly summarized below:

***Changes to Manuscript***

In Table 1, the model top for GEFSv12 has been corrected to 0.2 hPa, and an asterisk has been added next to its name.

In Figure 1, GEFSv12 has been moved toward the right side of the figure to be grouped with the low-top systems. The composite means have been updated accordingly, which has resulted in small changes to the composite global annual mean stratospheric temperatures listed in the numbers. The statistical significance of the differences between the high- and low-top composites did not change.

In Figure 2 (and the associated supplement Figure S15), GEFSv12 is now part of the low-top composite. This has resulted in minor changes to the appearance of the high-top composite biases, but the spatial patterns of biases remain consistent, and no conclusions are altered. It is also worth noting again that we have the biases for individual systems in the supplement (Figures S1-S14), and these remain unchanged.

In Figures 3, 4 (and the associated supplement Figures S16-S17) GEFSv12 is now plotted with dashed lines, and has asterisks next to its name in the legends to denote it is considered a low-top system.

In Figures 6 and 7, there are now asterisks next to GEFSv12 in the legends to denote it is considered a low-top system.

In the text, there were a few spots that referred to GEFSv12 as a high-top system which have been corrected. However, the substance of the statements did not change. E.g.:

*"However, it is still worth noting that at 10 hPa the magnitude of the low-top biases all exceed 1 K, while **all high-top systems except for GEFSv12 and NCEP** are below 1 K."* was changed to *"However, it is still worth noting that at 10 hPa the magnitude of the low-top biases all exceed 1 K, while **all high-top systems except for NCEP** are below 1 K."*

*"The **biases for the high-top systems** are generally smaller in magnitude, but there are some exceptions: for instance, **the CESM2-WACCM, ECMWF, GEFSv12, and NCEP systems** have biases at some levels that are as large or larger in magnitude than the low-top systems."* was changed to *"The biases for the high-top systems are generally smaller in magnitude, but there are some exceptions: for instance, **the CESM2-WACCM, ECMWF, and NCEP** systems have biases at some levels that are as large or larger in magnitude than the low-top systems."*

*"Several of the systems with the largest global- and annual-mean warm biases in the stratosphere are those in the ``NOAA family", including the **high-top GEFSv12 and NCEP CFSv2, and the low-top GFDL-SPEAR**"* was changed to *"Several of the systems with the largest global- and annual-mean warm biases in the stratosphere are those in the ``NOAA family", including the **high-top NCEP CFSv2, and the low-top GEFSv12 and GFDL-SPEAR**"*

I apologize for the error and any confusion it may have caused.

Sincerely,
ZDL

**References**
Hamill, T. M., Whitaker, J. S., Shlyaeva, A., Bates, G., Fredrick, S., Pegion, P., Sinsky, E., Zhu, Y., Tallapragada, V., Guan, H., Zhou, X., & Woollen, J. (2022). The Reanalysis for the Global Ensemble Forecast System, Version 12, Monthly Weather Review, 150(1), 59-79.
https://journals.ametsoc.org/view/journals/mwre/150/1/MWR-D-21-0023.1.xml